

# Aerosol-type classification based on AERONET version 3 inversion products

Sung-Kyun Shin[1], Matthias Tesche[2], Youngmin Noh[3], and Detlef Müller[1]

[1]School of Physics, Astronomy and Mathematics, University of Hertfordshire, Hatfield, United Kingdom
[2]Leipzig Institute for Meteorology, Leipzig University, Leipzig, Germany
[3]Department of Environmental Engineering, Pukyong National University, Busan, Republic of Korea

**Correspondence:** Youngmin Noh (nym@pknu.ac.kr)

**Abstract.** This study proposes an aerosol-type classification based on the particle linear depolarization ratio (PLDR) and single scattering albedo (SSA) provided in the AErosol RObotic NETwork (AERONET) version 3 level 2.0 inversion product. We compare our aerosol-type classification with an earlier method that uses fine-mode fraction (FMF) and SSA. Our new method allows for a refined classification of mineral dust that occurs as a mixture with other absorbing aerosols: pure dust (PD), dust-dominated mixed plume (DDM), and pollutant-dominated mixed plume (PDM). We test the aerosol classification at AERONET sites in East Asia that are frequently affected by mixtures of Asian dust and biomass-burning smoke or anthropogenic pollution. We find that East Asia is strongly affected by pollution particles with high occurrence frequencies of 50% to 67%. The distribution and types of pollution particles vary with location and season. The frequency of PD and dusty aerosol mixture (DDM+PDM) is slightly lower (34% to 49%) than pollution-dominated mixtures. Pure dust particles have been detected in only 1% of observations. This suggests that East Asian dust plumes generally exist in a mixture with pollution aerosols rather than in pure form. In this study, we have also considered data from selected AERONET sites that are representative of anthropogenic pollution, biomass-burning smoke, and mineral dust. We find that average aerosol properties obtained for aerosol types in our PLDR-SSA-based classification agree reasonably well with those obtained at AERONET sites representative for different aerosol types.

## 1 Introduction

Atmospheric aerosol particles play an essential role in the global climate system by affecting the Earth's radiation budget (*Stocker et al.*, 2013). Aerosol particles directly interact with solar and terrestrial radiation through absorption and scattering of radiation. Aerosols also can act as cloud condensation nuclei and ice nucleating particles whereby they alter cloud properties (i.e., cloud albedo, cloud lifetime) and lead to a change in precipitation efficiency (*Twomey*, 1974). Radiative forcing of aerosols is a useful parameter in quantifying the radiative effect of aerosols on climate change. Estimates of aerosol radiative forcing require information on the amount of aerosol loading as well as aerosol properties (*Bellouin et al.*, 2013). The sign and the magnitude of the aerosol radiative forcing depends on aerosol characteristics (*Stocker et al.*, 2013). Despite an unprecedented global coverage of atmospheric aerosol information, it is still a challenging task to assess the aerosol radiative effect accurately. Atmospheric aerosols are difficult to characterize in time and space due to their short lifetime and geographically diverse




sources and production mechanisms. Moreover, atmospheric aerosol particles from different natural and anthropogenic sources also frequently mix and undergo aging processes during transport, which influences optical and microphysical properties of the bulk aerosols. An improved discrimination of different aerosol types by observations, particularly from space, increases the accuracy of estimates of the aerosol radiative impact and is important for the assessment of aerosol dispersion modelling

(*Satheesh and Moorthy*, 2005).

Aerosol properties inferred from remote sensing measurements can be used to classify different aerosol types. Aerosol optical depth (AOD) and its spectral dependence which can be characterized by the Ångström exponent (AE) are typically used for aerosol classification from passive remote sensing observations. These parameters, together with the fine or coarse mode fraction of the aerosol size distribution, allow for a quantification of aerosol loading and aerosol size (*Schuster et al.*,

2006). Dominant aerosol types can then be inferred by using additional information on, e.g. possible source regions or transport pathways (*Boselli et al.*, 2012; *Shin et al.*, 2019). The use of spectral absorbing characteristics of aerosols add additional information on particle type (*Giles et al.*, 2012). For instance, *Kim et al.* (2007) used the fine mode fraction (FMF) from the MoDerate resolution Imaging Spectrometer (MODIS) to determine particle size and then used the aerosol index (AI) from the ozone monitoring instrument (OMI) to determine the light-absorbing characteristics of aerosols. Other studies have proposed

to use properties related to particle size and absorption to determine the dominant aerosol type, based on AErosol RObotic Network (AERONET, *Holben et al.* 1998) observations (*Lee et al.*, 2010; *Russell et al.*, 2010; *Giles et al.*, 2012). AERONET provides a global, long-term aerosol data set that includes spectral AOD, complex refractive index, particle size distribution, single scattering albedo (SSA), and absorption aerosol optical depth (AAOD). The information on particle size and light-absorbing properties enables the identification of major aerosol types such as mineral dust, anthropogenic pollution, biomass

burning smoke, and mixtures of aerosol particles. *Lee et al.* (2010) used SSA at 440 nm and FMF of AOD at 550 nm from AERONET to determine the absorbing properties and dominant size-mode of particles (i.e. fine or coarse mode) for aerosol classification. *Russell et al.* (2010) classified aerosol types based on the Absorption Ångström exponent (AAE). *Giles et al.* (2012) used SSA, AAE, AE, and FMF from AERONET to infer dominant aerosol types. More detailed and quantitative information useful for aerosol classification can be obtained from active aerosol remote sensing with lidar, particularly when

we include information on the particle linear depolarization ratio (PLDR or $\delta^p$) which is a very sensitive parameter with respect to particle shape (*Shin et al.*, 2019). Values of PLDR between 0.30 and 0.35 represent non-spherical particles (pure mineral dust, volcanic ash), while values close to zero indicate the presence of non-spherical particles. Values in between these two extreme values characterize mixtures of spherical and non-spherical particles. Measurements of PLDR can be used to estimate the contribution of these two fundamentally different particle types (*Freudenthaler et al.*, 2009; *Tesche et al.*, 2009;

*Burton et al.*, 2012, 2014, 2015; *Shin et al.*, 2015, 2019).

Several studies have determined aerosol types in which lidar-derived PLDR was combined with aerosol optical and microphysical properties inferred from AERONET observations (*Burton et al.*, 2012; *Russell et al.*, 2014). Although the combination of polarization lidar and AERONET observations might be an ideal option for aerosol classification, there are few AERONET stations that are equipped with this additional instrumentation.





Version 3 of the AERONET retrieval methodology provides us with spectral PLDR as standard inversion product. In this study, we use the information on the shape (PLDR) and light-absorbing properties (SSA) of particles for refining aerosol typing that can be obtained from using standard AERONET measurements. We retrieve the dominant aerosol type at selected AERONET sites in East Asia and investigate their seasonal variation and optical properties.

We describe our methodology in Section 2. In Section 3, we present and discuss our results. A summary and conclusion of this work is presented in Section 4.

## 2   Data and methodology

### 2.1   Parameters

Polarization lidar allows us to infer PLDR from measurements of the particle backscatter coefficient ($\beta_\lambda^{\mathrm{p}}$) in different planes of
polarization compared to the plane of polarization of the emitted linearly polarized laser light:

$$\delta^{\mathrm{p}} = \frac{\beta^{\mathrm{p},\perp}}{\beta^{\mathrm{p},\|}}. \tag{1}$$

AERONET instruments measure direct solar radiation and sky radiation. The measurements are automatically analyzed using the AERONET inversion algorithm (*Dubovik et al.*, 2006). The retrieved aerosol products are stored in the AERONET data base (https://aeronet.gsfc.nasa.gov/, last access: 30 January 2019). In the AERONET retrieval, the elements $F_{11}(\lambda)$ and $F_{22}(\lambda)$
of the Müller scattering matrix (*Bohren and Huffman*, 1983) are computed from the particle size distribution and the complex refractive index that can be inferred from AERONET level 2 data products. The element $F_{11}(\lambda)$ is proportional to the flux of scattered light in the case of unpolarized incident light while $F_{22}(\lambda)$ strongly depends on the angular and spectral distribution of the radiative intensity (*Bohren and Huffman*, 1983). From the elements $F_{11}(\lambda)$ and $F_{22}(\lambda)$ at the scattering angle 180°, $\delta^{\mathrm{p}}$ is computed as

$$\delta_\lambda^{\mathrm{p}} = \frac{1 - F_{22}(\lambda, 180°)/F_{11}(\lambda, 180°)}{1 + F_{22}(\lambda, 180°)/F_{11}(\lambda, 180°)}. \tag{2}$$

*Noh et al.* (2017) report that PLDR from the AERONET inversion products shows high correlation with lidar-derived values. PLDR can be used to calculate the contribution of dust to the particle backscatter coefficient for an external aerosol mixture. *Shimizu et al.* (2004) and *Tesche et al.* (2009) define the dust ratio ($R_{\mathrm{d}}$) as:

$$R_{\mathrm{d}} = \frac{(\delta^{\mathrm{p}} - \delta_{\mathrm{nd}}^{\mathrm{p}})(1 + \delta_{\mathrm{d}}^{\mathrm{p}})}{(\delta_{\mathrm{d}}^{\mathrm{p}} - \delta_{\mathrm{nd}}^{\mathrm{p}})(1 + \delta^{\mathrm{p}})}. \tag{3}$$

Here, $\delta_{\mathrm{nd}}^{\mathrm{p}}$ and $\delta_{\mathrm{d}}^{\mathrm{p}}$ indicate the PLDR of non-dust and pure dust particles, respectively. The corresponding values can be determined from lidar or AERONET observations (*Burton et al.*, 2014; *Shin et al.*, 2018). *Shin et al.* (2018) recently discussed AERONET-retrieved PLDR from different source regions. The authors find that PLDRs at 870 and 1020 nm are likely to be the two most reliable quantities. The authors based this finding on comparison to literature values obtained from lidar observations of pure dust particles. We accordingly apply the aerosol classification procedure to AERONET measurements at 1020 nm. We





used values of $\delta_{\mathrm{nd}}^{\mathrm{p}} = 0.02$ and $\delta_{\mathrm{d}}^{\mathrm{p}} = 0.30$ for pure dust for Asian dust (*Shin et al.*, 2018). When PLDR was lower than $\delta_{\mathrm{nd}}^{\mathrm{p}}$ or higher than $\delta_{\mathrm{d}}^{\mathrm{p}}$, $R_{\mathrm{d}}$ was set to 0 or 1, respectively.

The spectral SSA is the ratio of the scattering and extinction of light by particles and is generally used to define their absorbing characteristics. In this study, we use SSA to distinguish between non-dust particles that are non-absorbing (NA),
weakly absorbing (WA), moderately absorbing (MA), and strongly absorbing (SA).

## 2.2   Data selection

We selected AERONET sites in East Asia to develop our methodology. These sites are in a region that is influenced by both natural and anthropogenic aerosols such as anthropogenic particles from fossil fuel combustion, biomass-burning smoke, and mineral dust (*Nakata et al.*, 2016). We combine the stations Beijing and Xianghe into what we denote as Greater Beijing.
The stations Seoul and Yonsei_University are considered representative for Seoul. Osaka is represented by the stations Osaka and Shirahama. We considered Gosan to represent regional background aerosols in East Asia as there are few anthropogenic sources near this site. The AERONET inversion is only performed for measurements with a 440-nm AOD larger than 0.4 (*Dubovik et al.*, 2006). For the aerosol type classification we use the AERONET level 2.0 version 3 inversion products of particle size distribution, PLDR, and SSA available as of February 2018. This means that the time series considered here cover
the time frame from the beginning of measurements at the respective sites up to between the end of 2016 and the middle of 2017.

We considered other AERONET stations to evaluate the results of our aerosol classification. These stations are generally considered to be representative for (1) anthropogenic particles: Goddard Space Flight Centre (GSFC, USA), Ispra (Italy), and Mexico City (Mexico), (2) biomass-burning smoke: Alta Floresta (Brazil), Mongu (Zambia), and Abracos Hill (Brazil), and (3)
Saharan dust: Capo Verde (Cape Verde), Banizoumbou (Niger), and Dakar (Senegal). Detailed information on the AERONET sites considered in this study is provided in Table 1.

## 2.3   Correlation between PLDR and other optical properties

To investigate whether PLDR can be used as the base parameter for aerosol type classification we analyzed how SSA and volume particle size distribution vary with respect to different intervals of PLDR. These parameters have been used for aerosol
type classification in previous studies (*Kim et al.*, 2007; *Lee et al.*, 2010). Figure 1 shows clearly distinguishable patterns of spectral SSA. The increase of SSA with increasing wavelength for PLDR$> 0.28$ is characteristic for mineral dust (*Giles et al.*, 2012). In contrast, the decrease of SSA with increasing wavelength for PLDR$< 0.1$ is in line with anthropogenic pollution and biomass-burning smoke (*Giles et al.*, 2012). In addition, SSA at 1020 nm is remarkably different compared to SSA at other wavelengths according to the PLDR. Values are in the range between 0.91 and 0.94 for PLDR$< 0.1$ and between 0.96 and 0.99
for high PLDRs. In contrast to the other wavelengths, these differences are very small at 440 nm. The composition of mineral dust often includes clay minerals or iron oxides that lead to strong light-absorption at short wavelengths (*Derimian et al.*, 2008). The main light-absorber of anthropogenic pollution is black carbon or soot, which exhibits light absorption throughout the entire solar spectrum (*Bergstrom et al.*, 2002). Aerosol particles which are categorized by PLDR into dust particles



and pollution particles thus are relatively low light-absorbing at 440 nm compared to their light-absorbing capacity at other wavelengths.

Figure 2 shows that the variation of the fine-mode and the coarse-mode part of the size distribution with respect to PLDR is also clearly distinguishable. The fine-mode particles dominate at lower PLDR and their contribution decreases with increas-
ing PLDR. Conversely, the contribution of the coarse-mode to the total volume concentration (CMFvc) increases as PLDR increases. CMFvc is less than 0.5 for PLDR$<0.06$ and higher than 0.9 for PLDR$>0.30$.

Values of PLDR$\geq 0.3$ indicate nearly pure dust particles (*Freudenthaler et al.*, 2009; *Shin et al.*, 2018), whereas PLDR$<0.1$ is representative for anthropogenic pollution or biomass-burning smoke (*Burton et al.*, 2012). External mixtures of mineral dust and non-spherical particles reveal PLDRs in the range from 0.08 to 0.30. Consequently, PLDR varies according to the dominant
aerosol type in the mixture (*Shin et al.*, 2015).

## 2.4 Aerosol classification method

Atmospheric aerosols are typically classified into four major types according to their size and light-absorbing property. For instance, *Lee et al.* (2010) used FMF and SSA from the AERONET version 2 level 2.0 inversion products to classify aerosols into black carbon (BC), mineral dust, non-absorbing (NA) particles, and mixed particles. In our aerosol classification method,
we use PLDR for obtaining information on the shape of the particles. We also use $R_\mathrm{d}$ to set a threshold of PLDR for aerosol classification with respect to the contribution of mineral dust to the aerosol mixture. SSA is used to distinguish absorbing from non-absorbing aerosols. The reason for using PLDR instead of FMF, which has been used, e.g. in *Schuster et al.* (2006) and *Lee et al.* (2010), is that the former study provides a clearer separation between non-spherical dust particles and rather spherical non-dust particles. *Lee et al.* (2010) used a threshold of FMF$<0.4$ to identify mineral dust. *Jones and Christopher*
(2007) reported annually mean values of MODIS FMF for mineral dust of $0.45\pm0.05$. *Shin et al.* (2019) compared coarse-mode AOD provided by AERONET to dust AOD retrieved with the use of PLDR and showed that the former tends to overestimate the contribution of mineral dust to AOD. As a consequence, FMF or fine-mode AOD, when used as proxy for identifying non-dust aerosols, would lead to a systematic underestimation of the contribution of non-dust particles to the total aerosol plume. Further benefits of using PLDR rather than FMF for aerosol type classification will be discussed later.
Figure 3 shows a flowchart of our aerosol classification method. The contribution of dust in the aerosol mixture is determined based on using a threshold for the PLDR. Previous studies reported values for Asian dust of 0.08 to 0.35 at 532 nm (*Murayama et al.*, 2004; *Shimizu et al.*, 2004; *Shin et al.*, 2015). Asian dust generally mixes with other aerosol types during long-range transport, which leads to a wide range in PLDR. *Shin et al.* (2015) used 0.08 at 532 nm as the threshold value to identify a contribution of mineral dust in mixed aerosol plumes. *Shimizu et al.* (2004) defined 0.1 at 532 nm as the threshold
value for the classification of mixed mineral dust. In this study, we define pollution particles to show PLDR$<0.06$ and mineral dust to show PLDR$>0.28$. Values between 0.06 and 0.28 are considered to represent mixtures of mineral dust and pollution aerosols. Depending on $R_\mathrm{d}$, we classified observations that fall into the range of PLDR between 0.06 and 0.28 as pollution-dominated mixed aerosol (PDM, $R_\mathrm{d}=0.17-0.53$) and dust-dominated mixed aerosol (DDM, $R_\mathrm{d}=0.53-0.89$). PDM is characterized by PLDR between 0.06 and 0.16 while DDM is identified for PLDR between 0.16 and 0.28. The light-absorbing





properties and size of PDM is likely to be closer to the characteristics of pollution particles than to DDM. In the same way, the light absorbing properties and size for DDM are more likely to resemble the characteristics of pure dust.

SSA at 440 nm is usually used to identify dust particles (*Kim et al.*, 2007; *Lee et al.*, 2010). Here, we use SSA at 1020 nm as we have already classified mineral dust and non-dust particles based on PLDR. This means that we can make use of

the better contrast that we find between black carbon and dust particles when using SSA at 1020 nm, and we can use this contrast for our threshold-based identification of different absorbing particle types and their mixtures. The SSA at 1020 nm of water soluble aerosols, including sulphates, is close to unity (*Hess et al.*, 1998), whereas SSA is close to zero for BC (0.07 at 1020 nm, *Haywood and Ramaswamy* 1998). The SSA values for aerosol mixtures (e.g., BC with sulphate) vary, depending on the relative humidity and mixing ratio (e.g., 0.91 at 70% RH and 0.5 BC/sulphate mixing ratio for an internal mixture)

(*Wang and Martin*, 2007). *Dubovik et al.* (2002) report on SSA at 1020 nm of 0.83-0.95, 0.78-0.91, 0.97-0.99, and 0.97 for urban/industrial pollutant, biomass burning, desert dust, and marine aerosol, respectively. We note that pollution particles which were classified by PLDR could consist of both biomass-burning aerosol and anthropogenic aerosols. The biomass-burning aerosol contains BC, while anthropogenic aerosol contains BC and/or NA. Therefore, we conclude that anthropogenic aerosols with higher SSA consist mostly of NA whereas particles with lower SSA indicate that NA is mixed with BC. For

this reason, an SSA threshold of 0.95 is used to identify NA and to mark the upper limit for SSA of anthropogenic pollution. Depending on 1020-nm SSA, absorbing aerosols are then further divided into strongly absorbing (SA, SSA$< 0.85$), moderately absorbing (MA, SSA$= 0.85 - 0.9$), and weakly absorbing (WA, SSA$= 0.90 - 0.95$).

## 3 Results

### 3.1 FMF versus PLDR for aerosol-type classification

Figure 4 shows a 2-d histogram distribution of the relation between FMF and PLDR. Desert dust predominately consists of coarse-mode particles, whereas combustion-produced particles are predominantly found in the FMF of particle size distributions. The PLDR is close to zero for spherical particles such as anthropogenic or smoke particles and increases with increasing particle non-sphericity. The dominant aerosol type in the aerosol mixture can be distinguished based on knowledge of FMF and PLDR. We find a strong negative correlation of $R^2 = 0.80$ for FMF versus PLDR. The use of PLDR allows us to separate

between pure dust (PD, sector A), dust-dominated mixture (DDM, sectors B, C, and E), pollution-dominated mixture (PDM, sectors D and E), and pollution (sector G). This leads to an overall more comprehensive aerosol classification with respect to dusty mixtures.

Figure 5 shows the mean volume particle size distribution and SSA of the seven sectors. The optical and microphysical properties of the aerosols in each sector are listed in Table 2. The volume particle size distribution and SSA for the two dust

sectors A and B are markedly different. Sector A shows a mean FMF of $0.31 \pm 0.18$ while this value increases to $0.41 \pm 0.17$ in sector B. Additionally, the SSA at 1020 nm in sector A is $0.90 \pm 0.26$ while SSA in Sector B drop to $0.88 \pm 0.25$. Although both Sectors C and D are classified as mixed aerosols, particle size distribution of Sector C is characterized by a more dominant coarse mode than particle size distribution in Sector D.





The difference in the values of optical and microphysical properties of aerosol in Sectors E, F, and G (pollution dominated) is even more pronounced. Values of FMF are $0.71 \pm 0.09$, $0.80 \pm 0.08$, and $0.90 \pm 0.05$, respectively. SSA is $0.88 \pm 0.26$, $0.87 \pm 0.21$, and $0.85 \pm 0.25$, respectively. The spectral dependence of SSA in Sector E is also distinguishable from that of SSA in Sectors F and G. SSA increases with increasing wavelength in sector E. Such a spectral behaviour is a characteristic feature

of mineral dust. The SSA decreases with increasing wavelength in Sector G. We find little spectral variation of SSA in Sector F. Sectors F and G are more likely to represent pollution particles, whereas Sector E more likely contains mixtures of pollution and mineral dust.

The volume particle size distribution and spectral SSA in Sector A (pure dust based on PLDR) is clearly different from the ones in the other sectors. Sector A contains coarse-mode-dominated particle size distributions (FMF= $0.31 \pm 0.18$) and shows

a spectral SSA that resembles the one of mineral dust. Similarly, the spectral characteristics of SSA of aerosols in Sectors B, C, and E, which are classified as dust dominant mixtures, are characteristic for dust particles. However, SSA at 1020 nm is lower (0.84-0.88) compared to SSA at 1020 nm in Sector A. SSA at 1020 nm varies around of 0.87 in Sectors D and F. These sectors were classified as pollution-dominated mixtures (based on PLDR). The volume size distributions for Sectors D and F show that both coarse- and fine-mode particles are present in these mixtures. Sector G resembles the characteristics of pollution particles

(high FMF and low SSA).

We conclude that the FMF might be too ambiguous a parameter to distinguish aerosol types in mixed dust-pollution plumes. In contrast, PLDR is a more reliable parameter for aerosol type classification in which mineral dust is part of the aerosol. Our method is particularly useful for observations in East Asia were mineral dust is almost always mixed with pollution. We therefore propose a refined categorization into pure dust, dust-dominated mixtures, pollution-dominated mixtures, and

pollution.

## 3.2 Characteristics of aerosol types over East Asia

We investigated the regional and seasonal characteristics of aerosol over East Asia. The rates of occurrences of aerosol types at each AERONET observation site are shown in Figure 6. Regardless of season, pollution particles (NA+WA+MA+SA) are detected persistently over East Asia with 50%-67% occurrence rate. Pollution particles are detected most frequently at

Seoul (67%). The lowest occurrence rate of pollution of 50% was found at Gosan. The seasonal variation and specific type of pollution differ for the different sites. Aerosols classified as more light absorbing (MA+SA), are detected most frequently at Greater Beijing (SA: 4%, MA: 15%) and Osaka (SA: 5%, MA: 15%) followed by Seoul (SA: 2%, MA: 9%) and Gosan (SA: 3%, MA: 4%). The most frequently detected aerosol types over Seoul were NA (24%) and WA (32%). Differences in the frequency of different types of pollution particles, i.e. the BC-dominated types MA and SA and WA and NA which contain

little or no BC, can be explained by the sources of the aerosols. The occurrence rate of MA and SA at Greater Beijing (19%) is larger than at Seoul (11%). China has become a major source of BC because of fossil fuel combustion stemming from industry, domestic heating, and cooking. The contribution of BC in pollution thus has increased. In contrast, water-soluble (hygroscopic) aerosol particles such as sulphate or nitrate might be dominant in the anthropogenic pollution plumes over Seoul.


The occurrence rate of pure and polluted dust (PD, DDM, and PDM) over East Asia is slightly lower (34%-49%) than that of dust-free pollution. PDM is the most frequently detected aerosol type (over East Asia) of all types that include dust, i.e. PD, DDM, and PDM. The occurrence rates of PDM were 28%, 28%, 33%, and 23% for Greater Beijing, Seoul, Gosan, and Osaka, respectively. In contrast, the occurrence rate for PD was around 1% at most sites. The highest occurrence rate for PD is found at Greater Beijing (1%). Central Asian dust usually has traveled over densely populated and highly industrialized areas of China before it is detected. That means, as the result of transport path and time dust has usually mixed with pollutants before being detected (*Sun et al.*, 2005; *Shin et al.*, 2015). The lowest occurrence rate for PD and relatively higher occurrence rate for PDM and DDM over East Asia corroborates this assumption, i.e. that dust particles over East Asia are typically mixed with anthropogenic pollution.

The seasonal variation of each aerosol type retrieved with our aerosol classification method corroborates the findings of *Xia et al.* (2007), *Guo et al.* (2011), and *Jung et al.* (2009). Overall, dust-containing aerosol types (PD, DDM, and PDM) have the highest occurrence rates during spring with as much as 82% of cases for Gosan in April. This is in line with the Asian dust season in spring (*Guo et al.*, 2011). PD is observed at all sites during spring but only with low occurrence rates of 1% to 4%. PD is also rarely observed during the other seasons. The occurrence rates of pollutant particles are consistently high except during spring. The distributions and types of pollutant particles are rather complex because of the variety of their source with respect to location and season. Higher rates of SA and MA are detected at most observation sites during winter, which is likely the result of increased combustion of coal for domestic heating. In contrast, the Gosan site is less influenced by strong light-absorbing particles during winter compared to the other observation sites as it is located away from highly industrialized areas. The higher SA and MA occurrence rates detected during summer at Osaka might be due to regional climate effects: a high oxidant level from local emissions, high temperature, and strong thermal insulation support the conversion of volatile organic compounds into secondary organic aerosols (SOA, *Sano et al.* 2016). The aged SOA could produce brown carbon (BrC) which is significantly light-absorbing (*Liu et al.*, 2016).

Another feature to note is the occurrence rate of NA and WA which increases during summer at all sites. Sulphate and nitrate are major contributors to the aerosol loading during summer over East Asia. Strong solar radiation, high ambient temperature, and high relative humidity during summer in East Asia enhance photochemical processes. These conditions may have been responsible for the higher summer concentrations of ammonium sulphate and nitrate (*Jung et al.*, 2009). The Asian monsoon is another possible reason for the increased frequency of occurrence of NA and WA during summer. Increased relative humidity and associated hygroscopic particle growth during monsoon season can cause changes in aerosol scattering properties (*Xia et al.*, 2007). Regardless of season, significant amounts of NA and WA are detected at the coastal sites Gosan and Osaka. The high valued of NA and WA might be the result of large contributions of maritime aerosol which is strongly light-scattering as well as highly hygroscopic.

### 3.3 Dominant aerosol types at representative aerosol source regions

In order to validate our method, we used it to determine the dominant aerosol types at well-characterized AERONET sites representative for anthropogenic pollution, biomass burning particles, and mineral dust (*Dubovik et al.*, 2002; *Giles et al.*,



2012; *Choi et al.*, 2016). Details on the selected sites together with the occurrence rates of different aerosol types are given in Table 3.

The dominant aerosol type at dust sites was PD with occurrence rates of 64% to 81%. PD was most frequently detected at Cape Verde with 81% occurrence rate. Additionally, notable occurrences rate for DDM (16% to 35%) and PDM (0% to 4%)

are also found at the dust sites. This suggests that AERONET sites in or close to source regions of mineral dust are frequently affected by local pollution or other aerosols from sources upwind. The highest occurrence rate of 35% for mixed dust types (DDM+PDM) was found at Dakar where local pollution has a much stronger effect on aerosol composition compared to other Saharan sites (*Petzold et al.*, 2011).

The sites for anthropogenic pollution show an extremely high occurrence rate of pollutants of 93% to 99%, though the oc-

currence rates of the individual sub-types vary from site to site. Higher occurrence rates of NA and WA were found at GSFC (34% and 47%) and Ispra (32% and 37%). Conversely, a higher occurrence rate of MA (34%) and SA (29%) was found at Mexico City, which is frequently affected by severe air pollution that mostly consists of fine-mode and carbonaceous aerosols (*Choi et al.*, 2016). *Choi et al.* (2016) used cluster analysis in combination with AERONET data and found that secondary inorganic particles dominate at GSFC. Carbonaceous aerosol has strong light-absorption properties, whereas secondary inorganic

aerosols such as sulphates predominantly scatter light (*Bond and Bergstrom*, 2006).

The frequency distribution of SA and MA are higher at the biomass-burning sites. Those sites are considered to be affected by mostly light-absorbing particles from combustion. A high occurrence rate of 96% of SA was found at Mongu. MA is the most frequently detected type at Alta Floresta (42%) and Abracos Hill (41%). BC is produced by flaming fires, i.e. at high temperatures. This type of BC production is prevalent at Mongu, whereas production of primary organic carbon (OC) at lower

temperature (smoldering fires) is dominant at Alta Floresta (*Reid et al.*, 2005; *Choi et al.*, 2016). The occurrence rates of NA (4%) and WA (34) at Alta Floresta and Abracos Hill are also distinguishable from the rates at Mongu where less than 0.5% were detected as NA or WA. The photochemical formation of secondary inorganic aerosols due to emissions from biomass burning seems to be much more frequent at Alta Floresta and Abracos Hill compared to Mongu (*Choi et al.*, 2016).

Figure 7, analogous to Figures 1 and 2, presents the mean spectral SSA and volume particle size distributions for the

AERONET sites that are mainly affected by anthropogenic pollution, biomass-burning smoke, and Saharan dust. The spectral behaviour of SSA at these sites strongly resembles the ones presented by *Dubovik et al.* (2002), *Eck et al.* (2009), *Giles et al.* (2012), and *Shin et al.* (2018). Dust particles exhibit strong light absorption at short wavelengths and lower absorption in the visible and near infrared wavelength region (*Shin et al.*, 2018). Fine-mode particles and hygroscopic particles such as sulphate show nearly neutral spectral dependence of SSA and overall stronger light-scattering, i.e. higher SSA (*Dubovik et al.*, 2002).

BC has the strongest light-absorption properties in the near-infrared wavelength region. OC and BrC exhibit stronger light-absorption properties at ultraviolet and visible wavelengths (*Eck et al.*, 2009).

The SSA at sites that represent anthropogenic pollution likely reflects the spectral features of SSA of fine-mode particles and hygroscopic particles. The value of SSA at Mexico City is lower than at other anthropogenic sites in the entire wavelengths. Mexico City is frequently influenced by air pollution that likely consists of more carbonaceous aerosols, whereas GSFC and

Ispra are affected by secondary inorganic particles (*Choi et al.*, 2016). Additionally, the spectral dependence of SSA at the





biomass-burning sites is similar to that of BC. We note that the values of SSA at the Mongu site are much lower than the values of SSA at the other biomass-burning sites in the entire wavelength. Various absorbing aerosols (e.g., BC, OC, BrC, or secondary inorganic aerosols) can be emitted from biomass burning. The lower SSA at Mongu might be the result of a higher fraction of BC compared to the other biomass-burning sites. Consequently, the spectral dependence of SSA at Alta Floresta and

Abracos Hill is more neutral with high values of SSA compared to Mongu. BC has the strongest light-absorption properties in the near-infrared wavelength region, while OC and BrC exhibit stronger light-absorption properties at ultraviolet and visible wavelengths (*Eck et al.*, 2009). The spectral dependence of SSA at the dust sites is representative of mineral dust particles.

Figure 7 also compares the spectral SSA at the AERONET sites to the spectral SSA of the aerosol types defined in our classification method (see Figure 3). The SSA for NA is higher than SSA at the anthropogenic sites at all wavelengths. The

SSA for WA is similar to values obtained at GSFC and Ispra. We conclude from this that NA – in view of the definition of aerosol type – consists to a largest degree of light-scattering particles. Similarly, WA represents aerosols that contain both light-scattering and light-absorbing particles with the light-scattering contribution being predominant.

Anthropogenic sites contain light-scattering particles and light-absorbing particles, which is reflected in SSA. The spectral SSA of MA is similar to the values found at Alta Floresta and Abracos Hill except at the short wavelength. Consequently,

MA is an aerosol mixture in which light-absorbing particles have a stronger impact. The absorbing properties are most likely related to the contribution of OC. Finally, the spectral SSA of SA resembles the findings at Mongu and suggests that the light-absorbing properties of SA are related to a strong contribution of BC. We note that the SSA of SA and Mongu are different. SA has been defined based on observations in East Asia where absorbing particles are likely present in complex mixtures. In contrast, BC is considered as the dominant aerosol at Mongu.

The spectral SSA of DDM and PD is similar to the one found at the dust sites. However, the SSA of DDM is slightly lower than SSA of PD and of the AERONET dust sites as a result of mixing with pollutant particles. Accordingly, the SSA of PDM are even lower than of PD, DDM, and AERONET dust sites as it is defined to feature a larger contribution of pollutants.

The volume size distributions show a dominance of coarse-mode particles at the dust sites. Coarse-mode particles also contribute strongly to the total volume size distribution for PD and DDM, whereas a lower contribution of coarse-mode particles

is found for PDM. This result is in line with an increased concentration of anthropogenic pollution or biomass-burning smoke which, in the PDM type, are typically considered to be fine-mode particles (*Eck et al.*, 1999).

Fine-mode particles contribute strongest to the total volume size distribution at biomass-burning sites. Additionally, the contribution of fine-mode and coarse-mode seems evenly distributed in the total volume size distribution for MA and SA. *Reid et al.* (1998) found from studies in rain forest regions of Brazil that fresh smoke particles are significantly smaller than

well-aged smoke particles. We believe that fresh smoke particles contribute significantly at the biomass-burning sites, whereas MA and SA detected over East Asia are affected not only by fine-mode particles but also a considerable amount of coarse-mode particles, in contrast to the source regions of biomass burning. The contributions of fine-mode and coarse-mode particles to the total volume size distributions are rather similar for the anthropogenic sites. However the contribution of fine-mode particles to the total volume size distributions is dominant for NA.



## 4  Summary and conclusions

We presented a methodology that is used for classifying the dominant types of aerosol particles in mixed aerosol plumes. We used PLDR and SSA that are provided in the AERONET version 3 level 2.0 inversion products. Dusty aerosol mixtures were separated according to pure dust, dust-dominated mixtures, pollution-dominated mixtures and pollution. The pollutants were

classified as non-absorbing, weakly absorbing, moderately absorbing and strongly absorbing particles based on using SSA at 1020 nm. The new aerosol typing method provides detailed information on aerosol mixtures that contain varying levels of mineral dust.

We tested our method at East Asian AERONET sites that are frequently affected by various aerosol type. We found that aerosols categorized as pollutant (NA+WA+MA+SA) are most frequently detected over East Asia. The detection rate was

50%-67%. The distribution and types of pollutant vary according to the location of sites and season. The frequency distribution for PD or dust-containing aerosol (DDM+PDM) plumes is lower than that of pollutants. Moreover, PD was detected in less than 1% of the observations at most sites over East Asia. This suggests that dust over East Asia is almost always mixed with other types of aerosol.

We compared the results of our aerosol-type classification to the aerosol properties obtained from selected AERONET sites

which are considered to be representative for different aerosol types, i.e. anthropogenic pollution, biomass-burning smoke and mineral dust. For anthropogenic pollution, we found that that MA and SA occur most frequently at Mexico City, whereas GSFC and Ispra are affected most strongly by NA and WA. The frequency distribution of SA and MA is higher at the biomass-burning sites. Those sites are affected by mostly light-absorbing particles from combustion. PD is the dominant aerosol at dust sites. From the comparison to representative AERONET sites, we conclude that the method we use to identify dominant aerosol

types over East Asia also leads to reasonable results at other sites. The proposed method has the potential to provide improved information on aerosol type in regions where various types of aerosol are frequently present in the form of complex mixtures – not only over East Asia but also elsewhere on the globe. Application of our aerosol-typing method to the global AERONET data base will provide useful information for the validation of chemical transport modelling as well as (active) spaceborne sensors that provide PLDR observations.

*Data availability.*  The data used in this work are freely available through the AERONET portal at http://aeronet.gsfc.gov/.

*Author contributions.*  SKS, MT, and YN had the idea for this study. SKS and MT performed the data analysis and prepared the figures. All authors contributed to the discussion of the findings and the preparation of the manuscript.

*Competing interests.*  The authors declare that no competing interests are present.



*Acknowledgements.* We thank the principal investigators and their staff for establishing and maintaining the AERONET sites used in this investigation. This work was supported by a National Research Foundation of Korea (NRF) grant funded by the Korean government (NRF-2018R1D1A3B07048047).





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





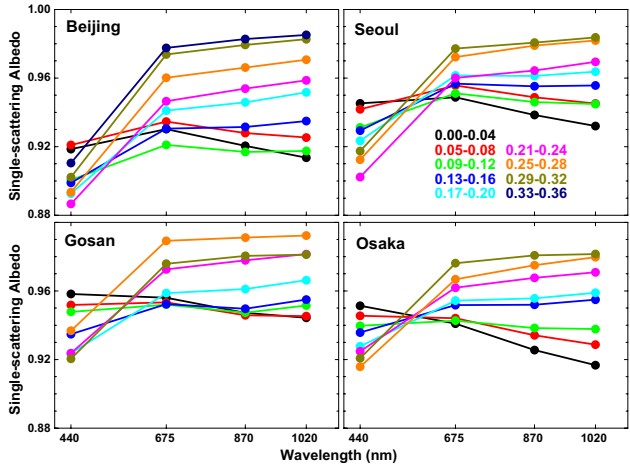

**Figure 1.** The mean spectra of SSA for different ranges of PLDR at the East Asian AERONET sites considered in this study.

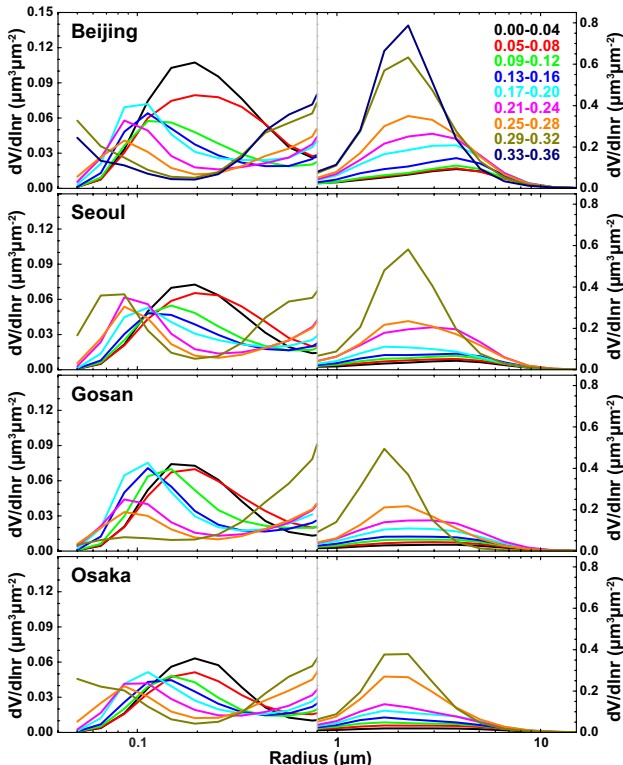

**Figure 2.** The mean volume particle size distributions for different ranges of PLDR for the East Asian AERONET sites considered in this study.





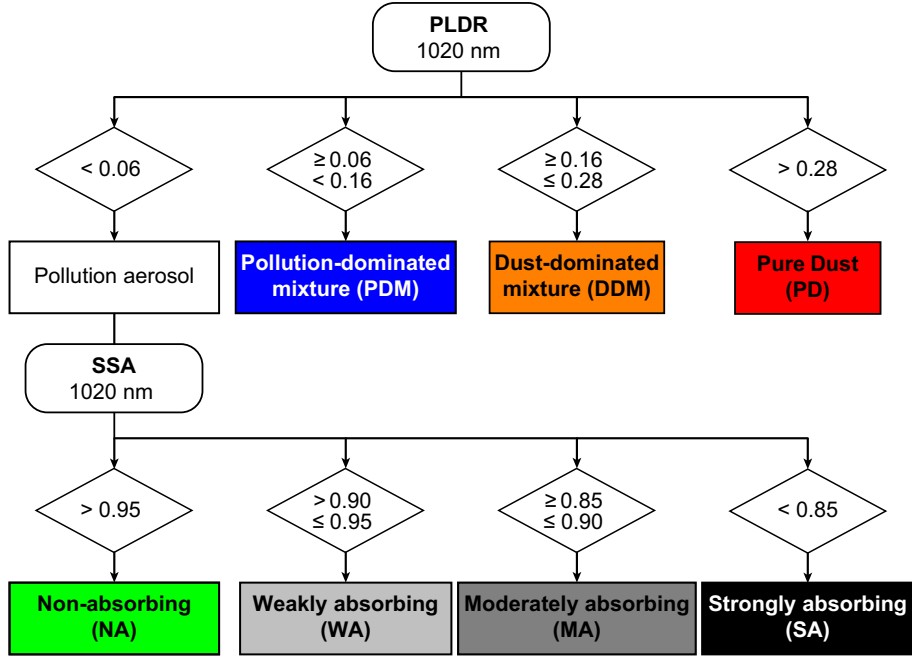

**Figure 3.** Flowchart of the aerosol classification based on the 1020-nm PLDR and the 1020-nm SSA that are inferred from the inversion of AERONET observations.

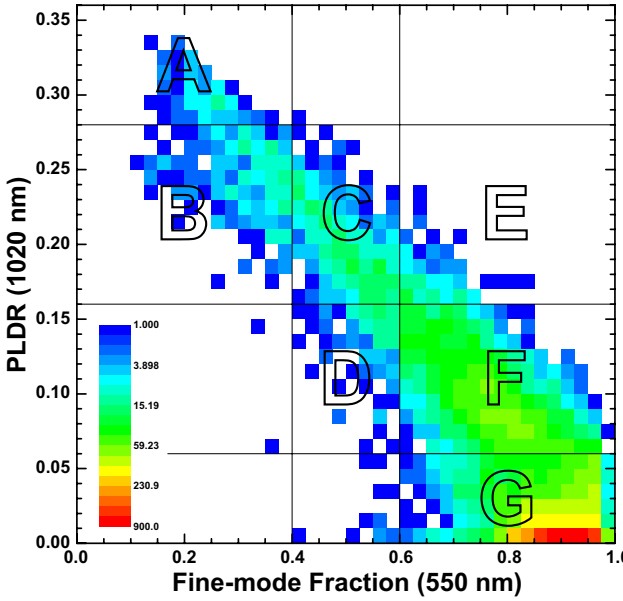

**Figure 4.** Two-dimensional histograms of 1020-nm PLDR and FMF for the considered AERONET stations over East Asia. Seven sectors A to G are defined according to the ranges of PLDR and FMF. The color coding indicates the number of observation data in log scale.





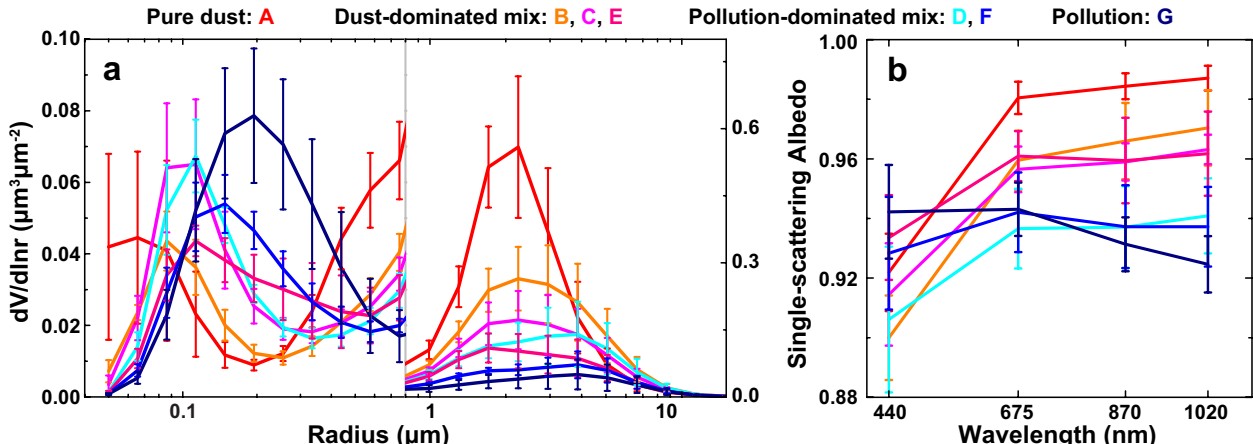

**Figure 5.** Mean volume size distributions (a) and spectral SSA (b) for the aerosol-type sectors identified in Figure 4.





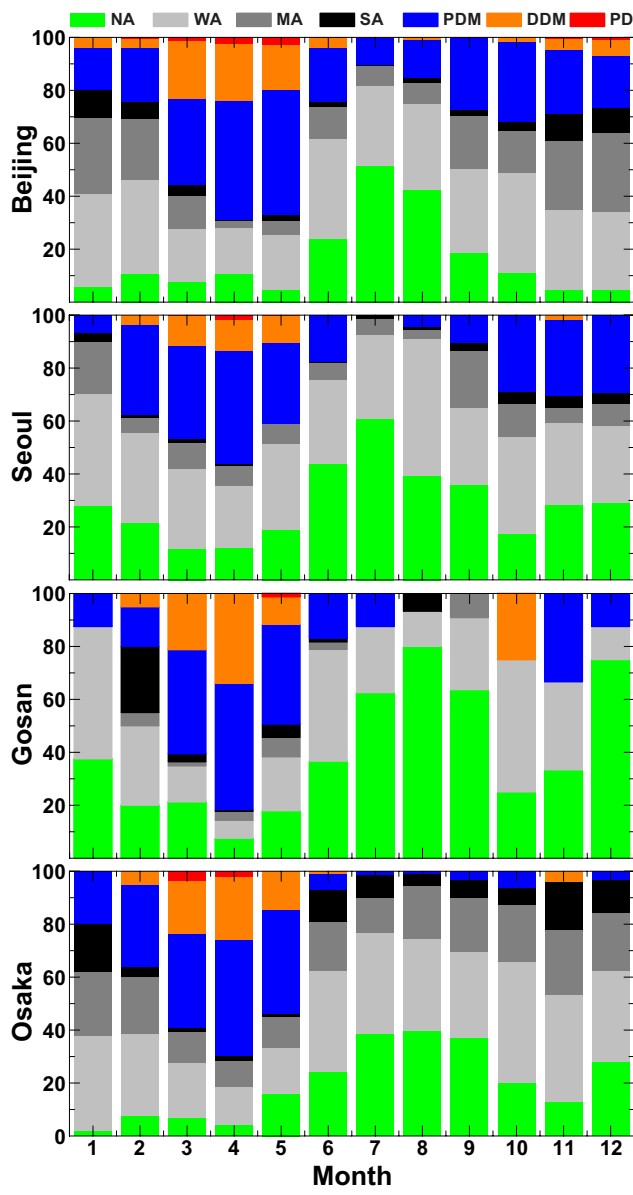

**Figure 6.** Occurrence rate of the aerosol types classified according to this study for AERONET observation sites over East Asia. The color coding refers to the aerosol species introduced in Figure 3.

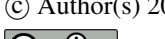



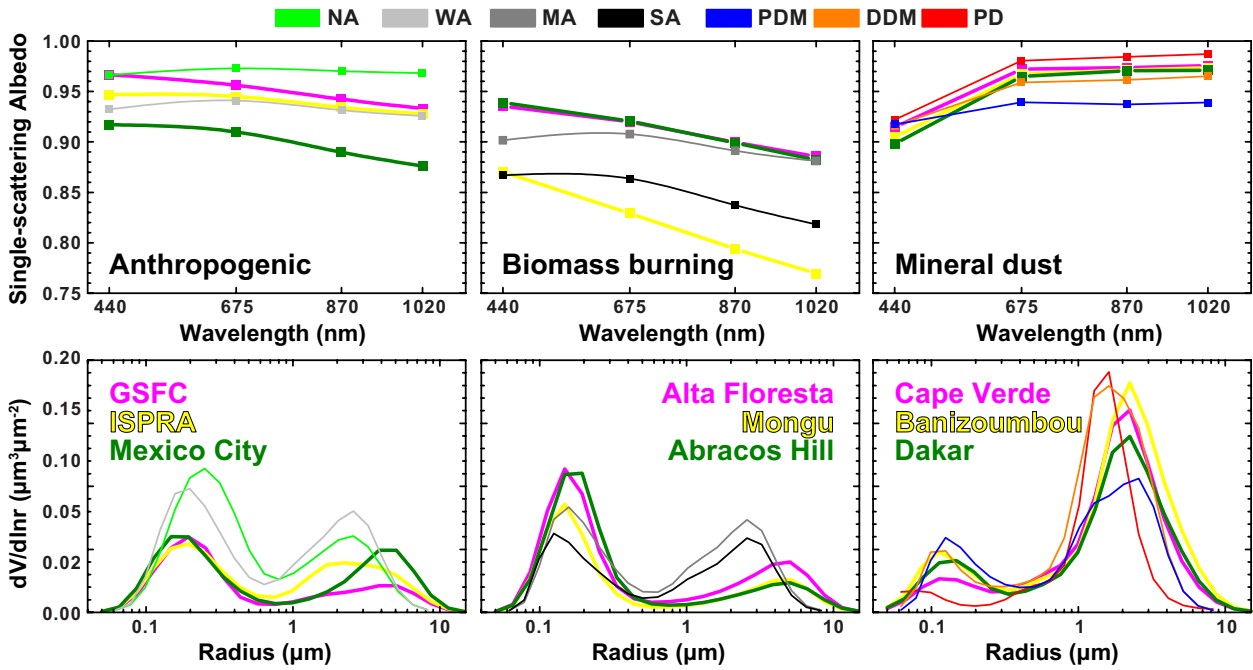

**Figure 7.** Mean spectral SSA (upper row) and mean particle size distributions (lower row) for AERONET sites that are considered as representative for different aerosol types (magenta, yellow and dark green) as well as for the aerosol types defined in this study (see Figure 3 for color coding). Note that the size distribution for pure mineral dust (red line) has been scaled to 50% to fit the plot.



**Table 1.** Geographical/retrieval information for each AERONET site used in this study, along with some references for each aerosol type.

| Site | Latitude [°] | Longitude [°] | Elevation [m] | Years | Cases |
|---|---|---|---|---|---|
| **Greater Beijing** | | | | | |
| Beijing | 39.98 | 116.38 | 92 | 2001-2017 | 4480 |
| Xianghe | 39.75 | 116.96 | 36 | 2001-2017 | 8289 |
| **Seoul** | | | | | |
| Seoul_SNU | 37.45 | 126.95 | 116 | 2002-2017 | 1715 |
| Yonsei_University | 37.56 | 126.94 | 97 | 2011-2017 | 2784 |
| **Gosan** | 33.29 | 126.16 | 72 | 2001-2016 | 1292 |
| **Osaka** | | | | | |
| Osaka | 34.65 | 135.59 | 50 | 2000-2016 | 2392 |
| Shirahama | 33.69 | 135.36 | 10 | 2000-2016 | 4080 |
| **Dust** | | | | | |
| Banizoumbou | 13.55 | 2.67 | 274 | 1995-2018 | 4217 |
| Capo_Verde | 16.73 | -22.94 | 60 | 1994-2018 | 1612 |
| Dakar | 14.39 | -16.96 | 21 | 1996-2018 | 3676 |
| **Biomass burning** | | | | | |
| Abrascos Hill | -10.76 | -62.36 | 259 | 1999-2015 | 392 |
| Mongu | -15.25 | 23.15 | 1047 | 1995-2010 | 1276 |
| Alta Floresta | -9.87 | -56.10 | 277 | 1993-2018 | 817 |
| **Anthropogenic** | | | | | |
| GSFC | 38.99 | -76.84 | 87 | 1993-2018 | 1237 |
| Ispra | 45.80 | 8.63 | 235 | 1997-2018 | 1114 |
| Mexico City | 19.33 | -99.18 | 2268 | 1999-2017 | 692 |



**Table 2.** Average values of AERONET-derived aerosol parameters for the sectors identified in Figure 5: FMF, AOD-related Ångström exponent for the wavelength pair 440 and 870 nm (AE), PLDR at 1020 nm, SSA at 1020 nm, total volume concentration (VolC), effective radius (Reff), and the volume median radius (VMR). For comparison, *Lee et al.* (2010) classifies categories A and B as dust, categories C and D as mixed, and categories E, F, and G as pollution.

| | A | B | C | D | E | F | G |
|---|---|---|---|---|---|---|---|
| | pure dust (PD) | dust domi-nated (DDM) | dust domi-nated (DDM) | pollutant domi-nated (PDM) | dust domi-nated (DDM) | pollutant domi-nated (PDM) | pollution |
| N | 90 | 355 | 520 | 207 | 512 | 3488 | 6691 |
| FMF | 0.31±0.18 | 0.41±0.17 | 0.59±0.13 | 0.62±0.09 | 0.71±0.09 | 0.80±0.08 | 0.90±0.05 |
| AE | 0.50±0.45 | 0.41±0.22 | 0.46±0.22 | 0.52±0.28 | 0.56±0.33 | 0.60±0.50 | 0.65±0.61 |
| PLDR | 0.28±0.04 | 0.22±0.03 | 0.17±0.03 | 0.13±0.02 | 0.16±0.02 | 0.09±0.02 | 0.02±0.01 |
| SSA | 0.90±0.26 | 0.88±0.25 | 0.84±0.31 | 0.87±0.22 | 0.88±0.26 | 0.87±0.21 | 0.85±0.25 |
| VolC | 0.52±0.31 | 0.44±0.27 | 0.38±0.18 | 0.40±0.15 | 0.38±0.21 | 0.40±0.23 | 0.50±0.30 |
| Reff | 0.73±0.36 | 0.59±0.33 | 0.40±0.20 | 0.43±0.19 | 0.37±0.15 | 0.30±0.12 | 0.28±0.11 |

**Table 3.** Occurrence rate (in percent and absolute numbers) of the aerosol types identified with our method at the selected AERONET sites.

| | cases | PD | DDM | PDM | NA | WA | MA | SA |
|---|---|---|---|---|---|---|---|---|
| **Anthropogenic** | | | | | | | | |
| GSFC | 1237 | 0 (0%) | 0 (0%) | 15 (1%) | 421 (34%) | 587 (47%) | 192 (16%) | 22 (2%) |
| Ispra | 1114 | 0 (0%) | 0 (0%) | 72 (6%) | 353 (32%) | 414 (37%) | 212 (19%) | 58 (5%) |
| Mexico City | 692 | 0 (0%) | 0 (0%) | 12 (2%) | 72 (10%) | 166 (24%) | 238 (34%) | 204 (29%) |
| **Biomass burning** | | | | | | | | |
| Alta Floresta | 817 | 0 (0%) | 0 (0%) | 6 (1%) | 31 (4%) | 275 (34%) | 341 (42%) | 164 (20%) |
| Mongu | 1276 | 0 (0%) | 0 (0%) | 0 (0%) | 4 (0%) | 3 (0%) | 50 (4%) | 1219 (96%) |
| Abracos Hill | 392 | 0 (0%) | 0 (0%) | 0 (0%) | 15 (4%) | 135 (34%) | 160 (41%) | 82 (21%) |
| **Saharan Dust** | | | | | | | | |
| Capo Verde | 1612 | 1311 (81%) | 260 (16%) | 29 (2%) | 1 (0%) | 0 (0%) | 3 (0%) | 7 (0%) |
| Banizoumbou | 4217 | 2749 (65%) | 1268 (30%) | 185 (4%) | 0 (0%) | 1 (0%) | 5 (0%) | 4 (0%) |
| Dakar | 3676 | 2204 (64%) | 1214 (35%) | 0 (0%) | 4 (0%) | 6 (0%) | 1 (0%) | 11 (0%) |