# Peer review of "Aerosol-type classification based on AERONET version 3 inversion products"

_Atmospheric Measurement Techniques, 2019_

## Referee Comment (RC1) · Anonymous Referee #2 · 9 May 2019

This paper attempts to determine the "aerosol-type" based upon the retrievals in the AERONET database. The main aerosol types defined in this paper are Pollution, Pollution-Dominated Mixtures (PDM), Dust-Dominated Mixtures (DDM), and Pure Dust (PD). Additionally, Pollution aerosols are divided further into four more subtypes, defined as Non-Absorbing (NA), Weakly Absorbing (WA), Moderately Absorbing (MA), and Strongly Absorbing (SA).

The main aerosol types are determined exclusively by the linear depolarization ratio (PLDR) at 1020 nm provided in the AERONET database. The pollution subtypes are determined exclusively by the single-scatter albedo (SSA) at 1020 nm.

The paper is well written and provides the reader with a relatively simple way of categorizing aerosols by type, but the paper would be much stronger if additional details

are added.

Major issues:

Page 2, Line 32: There are actually quite a few polarization lidars collocated with AERONET... see https://mplnet.gsfc.nasa.gov/ . All MPLNETs are collocated with AERONET, since the AOT constraint provided by AERONET is necessary for the MPLNET extinction profile retrievals. Granted, it is difficult (impossible?) to do meaningful statistical studies with the data because one can only download one day of MPLNET data at a time, but the hardware has been in place for many years.

Page 3, line 13: Dubovik (2006) is an article about incorporating spheroids into the retrieval, and is not the appropriate citation in this context. The authors should cite these papers instead:

Dubovik, O. and King, M.: A flexible inversion algorithm for retrieval of aerosol optical properties from sun and sky radiance measurements, J. Geophys. Res., 105, 20 673–20 696, 2000.

Dubovik, O., Smirnov, A., Holben, B., King, M., Kaufman, Y., Eck, T., and Slutsker, I.: Accuracy assess- ments of aerosol optical properties retrieved from Aerosol Robotic Network (AERONET) sun and sky radiance measurements, J. Geophys. Res., 105, 9791–9806, 2000.

Page 3, Equation 3: I don't understand why the authors present this equation in the section entitled "Parameters", as they never really use it and it never shows up in plots or the discussion. They do state in Section 2.4 (page 5, line 15) that they are using $R_d$ to define PLDR thresholds, and they provide a mapping between PLDR and $R_d$ on lines 32-35 of page 5. However, the definitions of Pollution, PDM, DDM, and dust are rather arbitrary for both parameters.

I would just remove $R_d$ from the paper altogether and define thresholds using the PLDR parameter. Alternatively, move Eq 3 into Sect 2.4 where there is some discussion of how this parameter is used, which would provide some context for the Equation... and then omit PLDR from the remainder of the paper and use R_d in the figures and discussion.

Page 3, line 26: "Shin et al (2018) recently discussed..." Some context needs to be provided here... why are PLDR(870) and PLDR(1020) more reliable than PLDR(440) or PLDR(675), when all of these PLDRS come from the same AERONET retrieval? I believe that the authors came to this conclusion because the AERONET PLDR values at visible wavelengths were biased low of lidar measurements at 532 nm, whereas the near-infrared AERONET PLDRS had higher values that were more consistent with the lidar values at 532 nm. Some discussion here would strengthen the choice of PLDR(1020).

Page 4, line 12 and elsewhere: There is not "Gosan" AERONET site... do the authors mean "Gosan SNU?"

Page 4, line 31: Authors state: "The composition of mineral dust often includes clay minerals or iron oxides that lead to strong light-absorption at short wavelengths"

Although iron oxides are highly absorbing, clay minerals are not.

Page 4, line 34: Authors state: "Aerosol particles which are categorized by PLDR into dust particles and pollution particles thus are relatively low light-absorbing at 440 nm compared to their light-absorbing capacity at other wavelengths."

I don't understand this logic... the authors just stated on lines 30-32 that "dust often contains... iron oxides that lead to strong light-absorption at short wavelengths." Also, authors mention biomass-burning on line 28, which also has strong light absorption at 440nm. So the authors contradict themselves.

Page 5, lines 3-6: Fig 2 does not show these concepts very well... the fine and coarse modes have different vertical axes, which makes it difficult to verify these words with the figure. There are also too many lines that are crossing all over the place, which

makes it hard to understand. Since you are really discussing CMFvc vs PLDR, why not just plot these parameters as a scatter plot?

Page 5, line 8-9: Authors state: "External mixtures of mineral dust and non-spherical particles reveal PLDRs in the range from 0.08 to 0.30." What is the basis of this statement? Was this shown in Burton 2012 or Shin 2015? As written, it does not make sense. Dust is non-spherical, so you're discussing mixtures of non-spherical dust with other non-spherical aerosols? It is important to be clear on this point because you are using PLDR to define thresholds for PDM and DDM. However, here you define dust mixtures as 0.08 < PLDR < 0.28, but everywhere else the lower threshold for PDM is 0.06 (not 0.08)... why the inconsistency?

Page 5, line 18: Authors state: "The reason for using PLDR instead of FMF, which has been used, e.g. in Schuster et al. (2006) and Lee et al. (2010), is that the former study provides a clearer separation between non-spherical dust particles and rather spherical non-dust particles."

This statement needs to be backed up with data, as it seems contrary to Figure 4 and page 6, line 24, which states: "We find a strong negative correlation of R^2 = 0.80 for FMF vs. PLDR."

Thus, the data presented in this paper indicates that both FMF and PLDR are capable of discriminating dust from pollution. No case has been made that one of these parameters is significantly better than the other.

Page 5, line 20: Authors state "Shin et al. (2019) compared coarse-mode AOD provided by AERONET to dust AOD retrieved with the use of PLDR and showed that the former tends to overestimate the contribution of mineral dust to AOD."

This sentence needs to be expanded, as it is unclear. First, what is being used as the "truth?" Are the authors assuming that their use of PLDR with the thresholds that they have chosen are the "truth?"

[Figure]

Page 6, line 9: Authors state: " The SSA values for aerosol mixtures (e.g., BC with sulphate) vary, depending on the relative humidity and mixing ratio (e.g., 0.91 at 70% RH and 0.5 BC/sulphate mixing ratio for an internal mixture) (Wang and Martin , 2007)."

Check this... SSA = 0.91 is way to high of a value for a 0.5 BC/sulphate mixing ratio and is inconsistent with Wang and Martin (2007), Figure 3.

Page 6, line 12: Authors state: "The biomass-burning aerosol contains BC, while anthropogenic aerosol contains BC and/or NA."

This sentence implies that biomass-burning (BB) aerosols do not contain non-absorbing aerosols (NA), which is not true. Also, BB aerosols often contain BrC that is highly absorbing at 440 nm but non-absorbing at 1020 nm.

Page 6, line 15: The setting of the "typing thresholds," both for SSA and PLDR, is the weakest element of this paper. Although the values chosen are reasonable (e.g., strongly absorbing aerosols have SSA < 0.85, Moderate absorbing have SSA 0.85-0.9, weakly absorbing SSA = 0.9-0.95, and non-absorbing have SSA > 0.95), they are basically pulled out of thin air.

Other studies (e.g., Burton 2012) use histograms and Mahalanobis distances to map multiple optical properties to known aerosol types.

Page 7, line 13: Authors state: "SSA at 1020 nm varies around of 0.87 in Sectors D and F."

This is inconsistent with Fig 5b, as SSA = 0.87 is not even on the scale (i.e., y-axis ranges from 0.88-1.0).

Page 7, lines 16-17: Authors state: "We conclude that the FMF might be too ambiguous a parameter to distinguish aerosol types in mixed dust-pollution plumes. In contrast, PLDR is a more reliable parameter for aerosol type classification in which mineral dust is part of the aerosol."

This is not at all obvious from the author's presentation. In fact, the argument is rather circular. i.e., use PLDR to distinguish type, then use FMF to validate conclusions, then discount FMF as ambiguous. Also, the authors note above that FMF is highly correlated with PLDR (R^2 = 0.8), so the two parameters are not really independent.

Page 7, lines 24-31: The authors need to add 12-month averages to Fig 6 so that the reader can follow along. None of the numbers in this paragraph correspond to the figure; presumably these numbers are averages over all 12 months, but the authors do not state this.

Page 8, line 33: This isn't really validation, as the authors do not have measurements to provide them with the "truth." Rather, this is basically a continuation of Section 3.2, but for different source regions.

I think the authors should include the seasonal variability at some of the sites in this section, similar to Fig 6. For instance, the West Africa has seasonal biomass burning as well as seasonal dust cycles. This should show up nicely at Banizoumbou and other African sites with a stacked chart like Fig 6.

Figure 4: Why show PLDR instead of R_p (from Eq 3)? If R_p really does present the contribution of dust to the particle backscatter coefficient, it would be much more interesting to see R_p than PLDR in this figure. Besides, PLDR only provides 4 broad categories of dust/pollution mixtures (dust, dust-dominated mixtures, pollution-dominated mixtures, and pollution), whereas the R_p would provide much higher resolution.

Figure 5: I still don't like the multi-scale y-axis. Maybe try a single log scale for y-axis, if a single linear scale does not work.

Colors are difficult to discriminate, even for the non- color blind. Color-blind people have no chance of understanding this figure. I would add symbols to the lines for those folks.

And you don't really need all those errorbars. . . May show one or more "representative"

errorbars.... or just state some representative values in the caption. What do the errorbars mean, anyways? 1-sigma? 2-sigma? 1 SDOM? 2 SDOM?

Table 2: Something wrong here... FMF ranges from 0.31 to 0.90, but AE only ranges from 0.5 to 0.65. AE = 0.52 and 0.60 for PDM regions D&F, and AE=0.65 for polluted region G. All of these values are inconsistent with the literature, where AE < 1 is often used for "pure" dust. At a minimum, the authors should provide some discussion as to why their polluted sector has such a low AE.

Table 3: This is a little messy to look at. You don't really need the number of cases for each aerosol type, since that is covered by the first numerical column and the percentages. If you get rid of the number of cases, you'll have one value (the percentage) for each aerosol type, and this will be a lot easier for the reader to process.

Alternatively, this table could be converted into a stacked bar chart of percentages (like Fig 6). One stacked bar for each site.

Minor issues:

Page 2, line 28: replace "non-spherical" with "spherical."

―――――――――――――――

---

## Referee Comment (RC2) · Patrick Hamill (Referee) · 14 May 2019

Review of ms for ACPD

Review of Paper by Shin, Tesche, Noh and Muller, ``Aerosol-type classification based on Aeronet version 3 inversion products"

In this paper, Shin et al. use some of the Aeronet inversion products to characterize aerosol types and to "investigate their seasonal and optical properties." They apply their method to four East Asian cities and they end up with 9 different aerosol types that differ from each other in one or another of the various Aeronet parameters. Whether or not the attempt is successful, I will leave to the reader to decide. But the paper does shed some interesting light on the problem of identifying aerosol types and I am recommending it be published after some corrections.

The paper focuses primarily on the aerosol measured at four eastern Asian sites, namely, Beijing, Seoul, Gosan and Osaka. There are few anthropogenic sources near Gosan, so the authors believe it represents the background aerosol of the region. For comparison purposes, they also included in their analysis Aeronet measurements from three presumably polluted sites (GSFC, ISPRA, and Mexico City), from three sites dominated by biomass burning smoke (Alta Floresta, Mongu, and Abracos Hill) and from three sites dominated by mineral dust (Cape Verde, Banizoumbou, and Dakar).

The Aeronet properties used in their analysis are the single scattering albedo (SSA), the size distribution (dV/dlnr) (Actually, the identification is based on the fine mode fraction which they denote FMF). These are supplemented with particle linear depolarization ratio (PLDR). They point out that PLDR is a sensitive parameter with respect to particle shape and can be used to determine whether dust particles are present. The PLDR can be obtained from polarization lidar measurements, but also can be calculated from some of the Aeronet data products. The authors mention that the derived PLDR used in their aerosol classification procedure is based on the Aeronet measurements at 1020 nm wavelength. They state that, "Values of PLDR between 0.30 and 0.35 represent non-spherical particles … while values close to zero indicate the presence of non-spherical particles." (Page 2 line 26) Presumably one or the other of the "non-sphericals" should be "spherical". (On page 4 line 1 we find that PLDR of 0.02 corresponds to "non-dust" and 0.30 corresponds to "Asian dust.")

I do not understand why the author's are calculating PLDR since they seem to be using it primarily to determine the sphericity of the particles. But the Aeronet data set has sphericity as a derived quantity. Why not just use the Aeronet value? I would have expected the authors to compare their sphericity evaluation with that of Aeronet, or at least comment on why they are not using the Aeronet value.

The paper reaches the conclusion that the aerosol in East Asia is mainly due to anthropogenic pollution and that dust aerosols are almost always mixed with other types of aerosol particle.

The introduction is well written and sets the stage for the analysis that follows.

The authors use the three parameters, fine mode fraction, single scattering albedo and polarization to define the type of aerosol. The aerosol types that they find are

Pure dust
Dust Dominated Mixtures
Polution Dominated Mixtures
Non-absorbing Pollution
Weakly absorbing Pollution
Moderately absorbing Pollution
Strongly absorbing Pollution
Biomass-burning Smoke

(It almost seems that a new aerosol type is defined for each variation in the three parameters.)

One of my main problems with the paper is the plethora of acronyms that make the paper difficult to read. For example, we read on page 8 lines 1-3, "The occurrence rate of pure and polluted dust (PD, DDM and PDM) over East Asia is slightly lower (34% -49%) than that of dust-free pollution. PDM is the most frequently detected aerosol type of all types that include dust, i.e., PD, DDM and PDM. The occurrence rates of PDM were…" Some acronyms are introduced and never used again. Thus on page 8 line 21 we read, "…secondary organic aerosols (SOA, Sano et al, 2016). The aged SOA …" The  acronym SOA is not found anywhere else in the paper. The paper would be much easier to read and understand if some of the acronyms were spelled out. There is nothing wrong with using acronyms, but when they are piled one upon another, the reader begins to flounder. If the authors insist on using so many acronyms, perhaps they could supply the reader with a look-up table of acronyms.

Section 3 presents their results. Figure 4 shows the values of PLDR vs fine mode fraction. The figure is broken up into 7 regions. It would be very helpful in understanding this figure if the caption would indicate the type of aerosol each region indicates. This information is given in the body of the text in lines 25-27 of page 6, but should be in the caption as well. This is done for Figure 5 and should be done here.

The caption of Figure 4 states that, "The color coding indicates the number of observation data in log scale." The color scale goes from 1 to 900. I suppose this means the number of observations range from 1 to 900. It would seem that most of the "pixels" represent less than about 30 readings with many being less than 3.

Figure 5 shows the size distributions (dV/dlnr vs r) for the seven regions presented in Figure 4.  The color scheme makes it hard to figure out what line corresponds to which kind of aerosol. Why not use green and yellow rather than three shades of red?

The second panel of Figure 5 gives SSA vs wavelength. The text (page 6 line 31) states that SSA in Sector B is 0.88. This agrees with the value given in Table 2, but it does not agree with Figure 5 which shows the lowest value of SSA to be about 0.92.

Page 7 lines 8-15. Once again, the values cited do not agree with the SSA values plotted in Figure 5.

Figure 6 presents the fraction of aerosol types observed at each site as function of month. The figure would be more informative if it gave the results in terms of the number of measurement rather than as a percentage. Thus, for example, one cannot appreciate the significance of the statement (page 7 line 24) "Pollution particles are detected most frequently at Seoul (67%)." I would suggest presenting the information like this:

[Figure]

In general the figure captions are not very helpful. They should not just tell us what is being plotted, but also what one is expected to see in the plots.

Specific Criticisms

Page 4 lines 18-19: The authors use Mexico City as a source of anthropogenic particles, that is, urban industrial aerosols. But as pointed out by Carabali et al. (10.1016/j.atmosres.2017.04.035), the Mexico City aerosol is dominated by two different types of aerosol, biomass burning and urban/industrial, depending on the season. (The burning of biomass by farmers in the hills surrounding the city is a major source of particulates.) It is not evident that the authors used the correct time frame for their presentation of Mexico City aerosols in Table 1 and Figure 7.

Page 4 lines 24-26 and Figure 1. The authors state that the plots of SSA vs wavelength for different values of PLDR show "clearly distinguishable patterns." The figure is hard to read and the "pattern" is not really clearly distinguishable. The number of curves in each panel range from 7 to 9, making it hard to appreciate what the authors are trying to show. I would suggest that instead of 9 ranges of PLDR, the point could be made with 3 or 4 ranges. The authors say, "In addition, SSA at 1020 nm is remarkably different compared to SSA at other wavelengths according to the PLDR." (I think they mean

"depending on" rather than "according to.") They go on to say, "Values are in the range between 0.91 and 0.94 for PLDR < 0.1 and between 0.96 and 0.99 for high PLDRs." This does not seem to me to be a significant difference since SSA can range from around 0.8 to 0.99. Finally, the different lines are different colors (representing different ranges of PLDR) but some of the colors are hard to tell apart. There are two shades of red and what appear to be two shades of black.

There is one thing about Figure 1 that I find puzzling. Since the PLDR is used to evaluate the sphericity of the particles, it seems strange that it would be different at 440 nm than at, say, 1020 nm. Does this mean that the particles that respond most strongly at the shorter wavelengths have a different sphericity than the particles that respond most strongly at the larger wavelength? I would think that such an effect would be worth describing and explaining.

Page 5 lines 3-10 and Figure 2. The figure shows the size distributions for the four East Asian sites. The caption should mention that the left column shows the fine mode and the right column shows the coarse mode and the vertical scales are different. Once again, the figure has far too many lines drawn on it, making it hard to understand.

Minor Corrections

Page 1 Line 19, "Radiative forcing of aerosols" should read "aerosol forcing by aerosols."

Page 3 line 14," https://aeronet.gsfc.nasa.gov/.last access: 30 January2019 " I assume the words ".last access…2019" are a typo.

Page 4 lines 1 and 2: The authors define the "Dust Ratio" $R_d$ in terms of $\delta_{nd}$ and $\delta_d$. In equation (3) there is also a $\delta$ with no subscript which is undefined, unless it is the $\delta$ defined in equation (1). They then state that, "We used values of $\delta_{nd}$ =0.02 and $\delta_d$ = 0.30 for pure dust for Asian dust." I don't know what the authors mean by "pure dust for Asian dust." Perhaps they mean "for pure dust and for Asian dust."

Page 4 line 3, "The spectral SSA is the ratio of the scattering and extinction of light…" This is not the correct definition of SSA.

Page 5 line 18. The word "study" should be deleted.

Page 10 line 9, Referring to Figure 7. "The SSA for NA is higher than SSA at the anthropogenic sites at all wavelengths." According to Figure 7, this is not true at 440 nm. (The thin green curve and the thick green curve are not easily distinguishable.)

Page 10 line13. "Anthropogenic sites contain…" I think the authors meant to say "Biomass burning sites contain…"

Page 10 lines 16-17. "Finally, the spectral SSA of SA resembles the findings at Mongu...We note that the SSA of SA and Mongu are different." Isn't this a contradiction?

Page 10 line 27. "Fine-mode particles contribute strongest to the ..." should read,"Fine-mode particles contribute most strongly to the ..."

---

## Author Comment (AC1) · 25 Jun 2019

Interactive comments on "Aerosol-type classification based on AERONET version 3 inversion products" by Sung-Kyun Shin et al.

Referee comments are noted in black. Our replies are given in blue.

We would like to thank all Referees for their constructive comments. Please find our pointby-point replies below. We have also attached a revised version of the manuscript with all changes marked.

**Anonymous Referee #1**

Review of Paper by Shin, Tesche, Noh and Muller, "Aerosol-type classification based on Aeronet version 3 inversion products"

In this paper, Shin et al. use some of the AERONET inversion products to characterize aerosol types and to "investigate their seasonal and optical properties." They apply their method to four East Asian cities and they end up with 9 different aerosol types that differ from each other in one or another of the various AERONET parameters. Whether or not the attempt is successful, I will leave to the reader to decide. But the paper does shed some interesting light on the problem of identifying aerosol types and I am recommending it be published after some corrections.

**We thank the Referee for this positive assessment.**

The paper focuses primarily on the aerosol measured at four eastern Asian sites, namely, Beijing, Seoul, Gosan and Osaka. There are few anthropogenic sources near Gosan, so the authors believe it represents the background aerosol of the region. For comparison purposes, they also included in their analysis Aeronet measurements from three presumably polluted sites (GSFC, ISPRA, and Mexico City), from three sites dominated by biomass burning smoke (Alta Floresta, Mongu, and Abracos Hill) and from three sites dominated by mineral dust (Cape Verde, Banizoumbou, and Dakar).

The AERONET properties used in their analysis are the single scattering albedo (SSA), the size distribution (dV/dlnr) (Actually, the identification is based on the fine mode fraction which they denote FMF). These are supplemented with particle linear depolarization ratio (PLDR). They point out that PLDR is a sensitive parameter with respect to particle shape and can be used to determine whether dust particles are present. The PLDR can be obtained from polarization lidar measurements, but also can be calculated from some of the AERONET data products. The authors mention that the derived PLDR used in their aerosol classification procedure is based on the AERONET measurements at 1020 nm wavelength. They state that, "Values of PLDR between 0.30 and 0.35 represent non-spherical particles ... while values close to zero indicate the presence of non-spherical particles." (Page 2 line 26) Presumably one or the other of the "non-sphericals" should be "spherical". (On page 4 line 1 we find that PLDR of 0.02 corresponds to "non-dust" and 0.30 corresponds to "Asian dust.").

**The Referee is right. We have corrected the typo. PLDR values close to zero refer to spherical particles.**

I do not understand why the authors' are calculating PLDR since they seem to be using it primarily to determine the sphericity of the particles. But the AERONET data set has

sphericity as a derived quantity. Why not just use the AERONET value? I would have expected the authors to compare their sphericity evaluation with that of AERONET, or at least comment on why they are not using the AERONET value.

We would like to elaborate on our rationale. Firstly, we didn't calculate PLDRs. This parameter is now included in the AERONET version 3 level 2.0 inversion product. Secondly, sphericity is no longer provided as a parameter in the AERONET version 3 level 2.0 inversion product. And thirdly, PLDR is physically more meaningful than sphericity as it is a parameter that can also be provided from independent measurements, i.e. with aerosol lidar. In particular, we are applying insights from lidar measurements of PLDR to refine AERONET-based aerosol typing. We have added the following statement to Section 2.1 for clarification:

"AERONET version 3 level 2.0 inversion products include PLDRs and lidar ratios at 440, 670, 870, and 1020 nm. The parameter sphericity, which was provided in the AERONET version 2 inversion product, has been discontinued in version 3. We hence base this study on PLDR. This approach has the additional advantage that PLDR can also be obtained from independent measurements, e.g. with aerosol lidar."

The paper reaches the conclusion that the aerosol in East Asia is mainly due to anthropogenic pollution and that dust aerosols are almost always mixed with other types of aerosol particle.

The introduction is well written and sets the stage for the analysis that follows. The authors use the three parameters, fine mode fraction, single scattering albedo and polarization to define the type of aerosol. The aerosol types that they find are: Pure dust, Dust Dominated Mixtures, Pollution Dominated Mixtures, Non-absorbing Pollution, Weakly absorbing Pollution, Moderately absorbing Pollution, Strongly absorbing Pollution, Biomass-burning Smoke

(It almost seems that a new aerosol type is defined for each variation in the three parameters.)

One of my main problems with the paper is the plethora of acronyms that make the paper difficult to read. For example, we read on page 8 lines 1-3, "The occurrence rate of pure and polluted dust (PD, DDM and PDM) over East Asia is slightly lower (34% -49%) than that of dust-free pollution. PDM is the most frequently detected aerosol type of all types that include dust, i.e., PD, DDM and PDM. The occurrence rates of PDM were..." Some acronyms are introduced and never used again. Thus on page 8 line 21 we read, "...secondary organic aerosols (SOA, Sano et al, 2016). The aged SOA ..." The acronym SOA is not found anywhere else in the paper. The paper would be much easier to read and understand if some of the acronyms were spelled out. There is nothing wrong with using acronyms, but when they are piled one upon another, the reader begins to flounder. If the authors insist on using so many acronyms, perhaps they could supply the reader with a look-up table of acronyms.

We understand the Referee's concern that our use of acronyms might get in the way of enabling a pleasant reading experience. We have tried to reduce the use of acronyms as much as possible. We are no longer using acronyms for secondary organic aerosols (SOA), Absorption Angstrom exponent (AAE), Angstrom exponent (AE), brown carbon (BrC), organic carbon (OC), coarse-mode to the total volume concentration (CMFvc), absorption aerosol optical depth (AAOD).

Section 3 presents their results. Figure 4 shows the values of PLDR vs fine mode fraction. The figure is broken up into 7 regions. It would be very helpful in understanding this figure if

the caption would indicate the type of aerosol each region indicates. This information is given in the body of the text in lines 25-27 of page 6, but should be in the caption as well. This is done for Figure 5 and should be done here.

A corresponding statement has been added to the figure caption.

The caption of Figure 4 states that, "The color coding indicates the number of observation data in log scale." The color scale goes from 1 to 900. I suppose this means the number of observations range from 1 to 900. It would seem that most of the "pixels" represent less than about 30 readings with many being less than 3.

It means that the highest found number of cases on a PLDR-FMF bin is 900 (red in the plot). The Referee is correct that many "pixels" contain very few cases. This is also reflected in the number of observations N in Table 2.

Figure 5 shows the size distributions (dV/dlnr vs r) for the seven regions presented in Figure 4. The color scheme makes it hard to figure out what line corresponds to which kind of aerosol. Why not use green and yellow rather than three shades of red?

We have revised Figure 5 for better contrast. We now only use a single y-scale. We have also changed the plot to black and gray lines as well as open and solid symbols to enable a better access to the figure also for colour blind readers.

The second panel of Figure 5 gives SSA vs wavelength. The text (page 6 line 31) states that SSA in Sector B is 0.88. This agrees with the value given in Table 2, but it does not agree with Figure 5 which shows the lowest value of SSA to be about 0.92.

We are sorry for this mistake. It turned out that the SSA values in Table 2 were not correct. We are now giving the correct numbers in Table 2 and in the mentioned part of the text. These numbers are now consistent with the plots in Figure 5.

Page 7 lines 8-15. Once again, the values cited do not agree with the SSA values plotted in Figure 5.

This has been a typo that we have now corrected to 0.94 in agreement with Figure 5b. See also our reply to the previous comment.

Figure 6 presents the fraction of aerosol types observed at each site as function of month. The figure would be more informative if it gave the results in terms of the number of measurement rather than as a percentage. Thus, for example, one cannot appreciate the significance of the statement (page 7 line 24) "Pollution particles are detected most frequently at Seoul (67%)." I would suggest presenting the information like this:

This has been a very good suggestion. We have revised Figure 6 to include the distribution of aerosol types of the entire year. We have added information of the number of considered observations per month and year to provide further context for the fraction of aerosol types in %.

In general the figure captions are not very helpful. They should not just tell us what is being plotted, but also what one is expected to see in the plots.

We respectfully disagree with the Referee. Figure captions need to state what is being plotted (i.e. enable understanding of the figure) while their description and interpretation belongs into the text.

**Specific Criticisms**

Page 4 lines 18-19: The authors use Mexico City as a source of anthropogenic particles, that is, urban industrial aerosols. But as pointed out by Carabali et al.

(10.1016/j.atmosres.2017.04.035), the Mexico City aerosol is dominated by two different types of aerosol, biomass burning and urban/industrial, depending on the season. (The burning of biomass by farmers in the hills surrounding the city is a major source of particulates.) It is not evident that the authors used the correct time frame for their presentation of Mexico City aerosols in Table 1 and Figure 7.

We thank the Referee for this comment. We did indeed miss to give a justification for our selection of sites. We have thus revised the beginning of the paragraph to: "We follow earlier AERONET studies on aerosol classification (Dubovik et al., 2001; Gobbi et al., 2007; Giles et al., 2012) to select stations that are generally considered to be representative for …"

We have looked at the paper by Carabali et al. (2017). They showed that the aerosol optical properties could be differed according to the season and it might be affected by different sources, i.e. anthropogenic or biomass burning. However, they conclude that the overall average of the Angstrom exponent (1.50) indicates the presence of fine particles at Mexico City. They also concluded that the bimodal structure of aerosol-size distribution at Mexico City resembles that found in some "other urban-industrial areas". The later indicates that anthropogenic sources dominate the local aerosol though it could be modulated by the presence of natural aerosols. They also concluded that aerosol absorption in Mexico City is close to AERONET retrievals in Beijing and Kanpur. This might be the reason why many studies treat the AERONET site Mexico City as urban-industrial.

Page 4 lines 24-26 and Figure 1. The authors state that the plots of SSA vs wavelength for different values of PLDR show "clearly distinguishable patterns." The figure is hard to read and the "pattern" is not really clearly distinguishable. The number of curves in each panel range from 7 to 9, making it hard to appreciate what the authors are trying to show. I would suggest that instead of 9 ranges of PLDR, the point could be made with 3 or 4 ranges. The authors say, "In addition, SSA at 1020 nm is remarkably different compared to SSA at other wavelengths according to the PLDR." (I think they mean "depending on" rather than "according to.") They go on to say, "Values are in the range between 0.91 and 0.94 for PLDR < 0.1 and between 0.96 and 0.99 for high PLDRs." This does not seem to me to be a significant difference since SSA can range from around 0.8 to 0.99. Finally, the different lines are different colors (representing different ranges of PLDR) but some of the colors are hard to tell apart. There are two shades of red and what appear to be two shades of black.

We have tidied up this figure. It now only shows SSA curves for four intervals of PLDR with clearly distinguishable colours. We have also revised the text based on the comments made by the two Referees.

There is one thing about Figure 1 that I find puzzling. Since the PLDR is used to evaluate the sphericity of the particles, it seems strange that it would be different at 440 nm than at, say, 1020 nm. Does this mean that the particles that respond most strongly at the shorter wavelengths have a different sphericity than the particles that respond most strongly at the larger wavelength? I would think that such an effect would be worth describing and explaining.

The literature on PLDR measurements at multiple wavelengths shows that the spectral behaviour of PLDR is indeed different for different aerosol types. This is because the effect depends on the size of the particles with respect to the measurement wavelengths. Particle shape becomes most important as particle size gets closer to the measurement wavelengths. Very non-spherical particles can basically be considered as spherical if they are very small compared to the used wavelength. We state in the text that we base our investigation on AERONET PLDRs at 1020 nm as those were found to closest resemble independent measurements with polarisation lidar.

Page 5 lines 3-10 and Figure 2. The figure shows the size distributions for the four East Asian sites. The caption should mention that the left column shows the fine mode and the right column shows the coarse mode and the vertical scales are different. Once again, the figure has far too many lines drawn on it, making it hard to understand.

We are sorry that this figure caused confusion. We have revised Figure 2 for better visibility. We now use the same vertical scale for fine and coarse mode. In addition, we only show four of the PLDR intervals in Figure 1. These are sufficient to visualise the change in the shape of the size distribution with PLDR. We hope that the revised figure is more suitable to transport the intended message.

**Minor Corrections**

Page 1 Line 19, "Radiative forcing of aerosols" should read "aerosol forcing by aerosols."

Changed to: "Radiative forcing is ..."

Page 3 line 14," https://aeronet.gsfc.nasa.gov/.last access: 30 January2019 " I assume the words ".last access...2019" are a typo.

According to Copernicus guidelines, links to websites have to be provided with an access date. We state the date when we had last accessed the AERONET website before submitting the manuscript. Nevertheless, we have changed this part to "*accessed on 30 January 2019*"

Page 4 lines 1 and 2: The authors define the "Dust Ratio" R d in terms of  $\delta$  nd and  $\delta$  d. In equation (3) there is also a  $\delta$  with no subscript which is undefined, unless it is the  $\delta$  defined in equation (1). They then state that, "We used values of  $\delta$  nd =0.02 and  $\delta$  d =0.30 for pure dust for Asian dust." I don't know what the authors mean by "pure dust for Asian dust." Perhaps they mean "for pure dust and for Asian dust."

We are indeed referring to the  $\delta$  defined in Eq. (1). The other part of this issue is a typo. The statement should have been "*for pure Asian dust*". This has been corrected.

Page 4 line 3, "The spectral SSA is the ratio of the scattering and extinction of light..."

This is not the correct definition of SSA.

To clarify, we have rephrased this statement to: "*The SSA is the ratio of the aerosol scattering coefficient to the aerosol extinction coefficient*"

Page 5 line 18. The word "study" should be deleted.

Done.

Page 10 line 9, Referring to Figure 7. "The SSA for NA is higher than SSA at the anthropogenic sites at all wavelengths." According to Figure 7, this is not true at 440 nm. (The thin green curve and the thick green curve are not easily distinguishable.)

The Referee is correct. The statement has been revised to: "*The SSA for NA is higher than SSA at the anthropogenic sites at all wavelengths except for the observation at GSFC at 440 nm*."

In addition, we have changed the colours used in this plot to provide better contrast between the different observations.

Page 10 line13. "Anthropogenic sites contain..." I think the authors meant to say "Biomass burning sites contain..."

**Corrected.**

Page 10 lines 16-17. "Finally, the spectral SSA of SA resembles the findings at Mongu...We note that the SSA of SA and Mongu are different." Isn't this a contradiction?

The Referee is correct. The statement has been changed to: "*Finally, the spectral SSA of SA is closest to the observations at Mongu*..."

Page 10 line 27. "Fine-mode particles contribute strongest to the ..." should read, "Fine-mode particles contribute most strongly to the ..."

We have changed the statement as recommended.

**Anonymous Referee #2**

This paper attempts to determine the "aerosol-type" based upon the retrievals in the AERONET database. The main aerosol types defined in this paper are Pollution, Pollution-Dominated Mixtures (PDM), Dust-Dominated Mixtures (DDM), and Pure Dust (PD). Additionally, Pollution aerosols are divided further into four more subtypes, defined as Non-Absorbing (NA), Weakly Absorbing (WA), Moderately Absorbing (MA), and Strongly Absorbing (SA).

The main aerosol types are determined exclusively by the linear depolarization ratio (PLDR) at 1020 nm provided in the AERONET database. The pollution subtypes are determined exclusively by the single-scatter albedo (SSA) at 1020 nm.

The paper is well written and provides the reader with a relatively simple way of categorizing aerosols by type, but the paper would be much stronger if additional details

We thank the Referee for this positive assessment.

**Major issues:**

Page 2, Line 32: There are actually quite a few polarization lidars collocated with AERONET... see https://mplnet.gsfc.nasa.gov/. All MPLNETs are collocated with AERONET, since the AOT constraint provided by AERONET is necessary for the MPLNET extinction profile retrievals. Granted, it is difficult (impossible?) to do meaningful statistical studies with the data because one can only download one day of MPLNET data at a time, but the hardware has been in place for many years.

Thank you for this comment. We have elaborated our statement by adding: ", *i.e. well-calibrated polarisation-sensitive micro-pulse lidar or aerosol polarisation lidar*"

Page 3, line 13: Dubovik (2006) is an article about incorporating spheroids into the retrieval, and is not the appropriate citation in this context. The authors should cite these papers instead:

Dubovik, O. and King, M.: A flexible inversion algorithm for retrieval of aerosol optical properties from sun and sky radiance measurements, J. Geophys. Res., 105, 20673-20696, 2000.

Dubovik, O., Smirnov, A., Holben, B., King, M., Kaufman, Y., Eck, T., and Slutsker, I.: Accuracy assessments of aerosol optical properties retrieved from Aerosol Robotic Network (AERONET) sun and sky radiance measurements, J. Geophys. Res., 105, 9791-9806, 2000.

**Thank you for the advice. The two papers have been added and cited here.**

Page 3, Equation 3: I don't understand why the authors present this equation in the section entitled "Parameters", as they never really use it and it never shows up in plots or the discussion. They do state in Section 2.4 (page 5, line 15) that they are using R\_d to define PLDR thresholds, and they provide a mapping between PLDR and R\_d on lines 32-35 of page 5. However, the definitions of Pollution, PDM, DDM, and dust are rather arbitrary for both parameters.

I would just remove  $R_d$  from the paper altogether and define thresholds using the PLDR parameter. Alternatively, move Eq 3 into Sect 2.4 where there is some discussion of how this parameter is used, which would provide some context for the Equation... and then omit PLDR from the remainder of the paper and use  $R_d$  in the figures and discussion.

These are very good recommendations that we have implemented in the revised manuscript. We kept Eq. (3) as it provides an easy connection between PLDR and the amount of non-spherical particles but we followed the Referee's suggestion and moved the description of  $R_d$ . However, we thought it better to place it at the end of Section 2.3 rather than into Section 2.4 as recommended. We have changed Figure 4 to show  $R_d$  over FMF and have changed the text to focus on  $R_d$  rather than PLDR as suggested by the Referee.

Page 3, line 26: "Shin et al (2018) recently discussed..." Some context needs to be provided here... why are PLDR(870) and PLDR(1020) more reliable than PLDR(440) or PLDR(675), when all of these PLDRS come from the same AERONET retrieval? I believe that the authors came to this conclusion because the AERONET PLDR values at visible wavelengths were biased low of lidar measurements at 532 nm, whereas the near-infrared AERONET PLDRS had higher values that were more consistent with the lidar values at 532 nm. Some discussion here would strengthen the choice of PLDR(1020).

We have added a corresponding statement: "The authors based this finding on comparison to literature values obtained from lidar observations of pure dust particles which shows that AERONET-derived PLDRs at lower wavelengths are systematically lower than lidar observations at 532 nm."

Page 4, line 12 and elsewhere: There is not "Gosan" AERONET site... do the authors mean "Gosan SNU?"

Yes we do. Gosan has been changed to Gosan\_SNU throughout the manuscript.

Page 4, line 31: Authors state: "The composition of mineral dust often includes clay minerals or iron oxides that lead to strong light-absorption at short wavelengths"

Although iron oxides are highly absorbing, clay minerals are not.

The reference to clay mineral has been removed.

Page 4, line 34: Authors state: "Aerosol particles which are categorized by PLDR into dust particles and pollution particles thus are relatively low light-absorbing at 440 nm compared to their light-absorbing capacity at other wavelengths."

I don't understand this logic... the authors just stated on lines 30-32 that "dust often contains... iron oxides that lead to strong light-absorption at short wavelengths." Also, authors mention biomass-burning on line 28, which also has strong light absorption at 440nm. So the authors contradict themselves.

We are sorry for the confusing and incorrect statement. The last sentence of this paragraph has not been correct. We have revised it to: "*Cases with considerable contribution of mineral dust as identified by high PLDR values therefore show relatively low SSA at 440nm compared to longer wavelengths.*"

Page 5, lines 3-6: Fig 2 does not show these concepts very well... the fine and coarse modes have different vertical axes, which makes it difficult to verify these words with the figure. There are also too many lines that are crossing all over the place, which makes it hard to understand. Since you are really discussing CMFvc vs PLDR, why not just plot these parameters as a scatter plot?

We agree that the figure has been too busy to understand easily. We have now changed it to show a single linear y-axis. We have also reduced the number of considered PLDR intervals. This makes for a tidier figure that still shows the change in size distribution for different intervals of PLDR, i.e. levels of dust contribution. We would like to keep the figure focussed on the actual size distributions as CMFvc vs PLDR would be very similar to FMF vs R\_d in figure 4.

Page 5, line 8-9: Authors state: "External mixtures of mineral dust and non-spherical particles reveal PLDRs in the range from 0.08 to 0.30." What is the basis of this statement? Was this shown in Burton 2012 or Shin 2015? As written, it does not make sense. Dust is non-spherical, so you're discussing mixtures of non-spherical dust with other non-spherical aerosols? It is important to be clear on this point because you are using PLDR to define thresholds for PDM and DDM. However, here you define dust mixtures as 0.08 < PLDR < 0.28, but everywhere else the lower threshold for PDM is 0.06 (not 0.08)... why the inconsistency?

The difference in the range is due to the different wavelength. The statement refers to lidar measurements at 532 nm. We now mention that the values refer to that wavelength. We have also added references regarding the basis of this statement, i.e. Tesche et al. (2009, 2011) and Gross et al. (2011).

Page 5, line 18: Authors state: "The reason for using PLDR instead of FMF, which has been used, e.g. in Schuster et al. (2006) and Lee et al. (2010), is that the former study provides a clearer separation between non-spherical dust particles and rather spherical non-dust particles."

This statement needs to be backed up with data, as it seems contrary to Figure 4 and page 6, line 24, which states: "We find a strong negative correlation of  $R^2 = 0.80$  for FMF vs. PLDR."

Thus, the data presented in this paper indicates that both FMF and PLDR are capable of discriminating dust from pollution. No case has been made that one of these parameters is significantly better than the other.

We agree that our statement has not been clear. We wanted to emphasize that PLDR and R\_d are sensitive to particle shape. This allows for a direct detection of non-spherical dust particles without having to assume that dust particles are confined to the coarse mode. FMF can only be used as a proxy for the contribution of mineral dust as long as the coarse mode is made up exclusively of dust particles. Figure 4 shows a large number of cases with FMF>0.4 (the threshold for mineral dust in *Lee et al.*, 2010) that reveal R\_d of 0.6 and higher. While mineral dust particles are generally considered as both large in size and non-spherical in shape, we believe that shape is a better indicator for mineral dust than size. This is also the rationale behind the statement the Referee addresses in the next comment, i.e. the reference to Shin et al. (2019) regarding the comparison of coarse-mode AOD and PLDR-derived dust AOD.

We have elaborated out statement and hope that the issue is now better addressed. We now state: "The reason for using  $R_d$  instead of FMF, which has been used, e.g. in Schuster et al. (2006) and Lee et al. (2010), is that the former provides a clearer separation between dust and non-dust particles. The sensitivity of  $R_d$  to particle shape allows for a straightforward and size independent identification of non-spherical dust particles in a mixture while it has to

**be assumed that dust is constrained to the coarse mode when using FMF for aerosol classification. However, dust might also be present in the fine mode (Mamouri and Ansmann, 2014, 2017)."**

Page 5, line 20: Authors state "Shin et al. (2019) compared coarse-mode AOD provided by AERONET to dust AOD retrieved with the use of PLDR and showed that the former tends to overestimate the contribution of mineral dust to AOD."

This sentence needs to be expanded, as it is unclear. First, what is being used as the "truth?" Are the authors assuming that their use of PLDR with the thresholds that they have chosen are the "truth?"

The statement refers to our earlier study in which we have adapted the PLDR-based aerosoltype separation of Tesche et al. (2009) to derive the contribution of mineral dust to AOD. The approach is based rather conservative PLDR thresholds for pure mineral dust and non-dust aerosols. We cannot know if this dust-related AOD is actually the truth. But we know that dust is not always confined exclusively to the coarse mode which is the basic assumption of using FMF as dust proxy. We can therefore state that separating the contributions of nonspherical and spherical particles based on parameters that are sensitive to particle shape is more meaningful for separating between dust, pollution, and their mixtures than using FMF. We hope that this rationale is now easier to understand based on the statement added in the reply to the previous comment.

Page 6, line 9: Authors state: "The SSA values for aerosol mixtures (e.g., BC with sulphate) vary, depending on the relative humidity and mixing ratio (e.g., 0.91 at 70% RH and 0.5 BC/sulphate mixing ratio for an internal mixture) (Wang and Martin, 2007)."

Check this... SSA = 0.91 is way too high of a value for a 0.5 BC/sulphate mixing ratio and is inconsistent with Wang and Martin (2007), Figure 3.

The Referee is correct. The mixing ratio we are referring to should have been 0.05 which corresponds to a SSA of 0.89 at 70% RH in Figure 3 of *Wang and Martin* (2007). The statement has been revised accordingly.

Page 6, line 12: Authors state: "The biomass-burning aerosol contains BC, while anthropogenic aerosol contains BC and/or NA."

This sentence implies that biomass-burning (BB) aerosols do not contain non-absorbing aerosols (NA), which is not true. Also, BB aerosols often contain BrC that is highly absorbing at 440 nm but non-absorbing at 1020 nm.

We are sorry for the confusing statement. We realised that it is not properly transporting the intended message. The statement has been revised to: "We note that spherical non-dust particles which were classified by PLDR could contain either biomass-burning aerosol, anthropogenic aerosols, or both. For simplicity and because AODs during purely marine conditions are generally too low to yield the AERONET level 2.0 inversion products used in this study, we refer to these particles as pollution. We now assume that pollution with higher SSA consist mostly of NA whereas particles with lower SSA indicate that NA is mixed with BC. For this reason, an SSA threshold of 0.95 is used to identify NA and to mark the upper limit for pollution aerosols. Depending on 1020-nm SSA, absorbing aerosols are then further divided into, and weakly absorbing (WA, SSA= 0.90-0.95), moderately absorbing (MA, SSA= 0.85-0.9), and strongly absorbing (SA, SSA< 0.85). For reference, biomass-burning aerosol

**generally contains a larger fraction of BC (and often brown carbon as well) compared to anthropogenic aerosol (Washenfelder et al., 2015), and thus, is likely to fall into the more absorbing categories."**

Page 6, line 15: The setting of the "typing thresholds," both for SSA and PLDR, is the weakest element of this paper. Although the values chosen are reasonable (e.g., strongly absorbing aerosols have SSA < 0.85, Moderate absorbing have SSA 0.85-0.9, weakly absorbing SSA = 0.9-0.95, and non-absorbing have SSA > 0.95), they are basically pulled out of thin air.

Other studies (e.g., Burton 2012) use histograms and Mahalanobis distances to map multiple optical properties to known aerosol types.

We are sorry for missing to provide reference to the origin of the threshold values. The ones for the dust ratio R\_d are based on the lidar literature on aerosol-type separation based on measurements of the particle linear depolarisation ratio. Values have been transformed into R\_d as suggested by the Referees. The threshold values of SSA have been adopted from the AERONET-based aerosol classification presented by *Lee et al.* (2010). For clarity, we have revised the statement indicated by the Referee to: "*Depending on 1020-nm SSA and adapting the threshold values of* Lee et al. (2010), *absorbing aerosols are then further divided into, and weakly absorbing* (WA, SSA = 0.90–0.95), moderately absorbing (MA, SSA = 0.85–0.9), and strongly absorbing (SA, SSA < 0.85)."

Page 7, line 13: Authors state: "SSA at 1020 nm varies around of 0.87 in Sectors D and F."

This is inconsistent with Fig 5b, as SSA = 0.87 is not even on the scale (i.e., y-axis ranges from 0.88-1.0).

Thank you for catching this mistake. It was a typo that has now been corrected to 0.94 in agreement with Figure 5b.

Page 7, lines 16-17: Authors state: "We conclude that the FMF might be too ambiguous a parameter to distinguish aerosol types in mixed dust-pollution plumes. In contrast, PLDR is a more reliable parameter for aerosol type classification in which mineral dust is part of the aerosol."

This is not at all obvious from the author's presentation. In fact, the argument is rather circular. i.e., use PLDR to distinguish type, then use FMF to validate conclusions, then discount FMF as ambiguous. Also, the authors note above that FMF is highly correlated with PLDR ( $R^2 = 0.8$ ), so the two parameters are not really independent.

We are sorry for the confusion caused by our line of reasoning. In the reply to earlier Referee comments, we have already elaborated on why we think that a parameter that is sensitive to particle shape (i.e. PLDR or R\_d) is better suited to identify non-spherical mineral dust particles than a parameter that is related to particle size, i.e. FMF. There is a strong correlation between the two parameters as mineral dust is predominantly coarse-mode aerosol. FMF is no longer a suitable proxy for dust if dust particles are also present in the fine mode. However, shape-dependent properties would still be sensitive to these particles as has been shown, e. g. by *Mamouri and Ansmann* (2014; 2017) which we now cite in the paper. For clarity, we have revised the statement to: "*As in our earlier study (Shin et al., 2019), we conclude that FMF with its relation to particle size might be too ambiguous a parameter to distinguish aerosol types in mixed dust-pollution plumes. In contrast, PLDR and R\_d are related to particle*

**shape, and thus, likely to be better suited and physically more meaningful for aerosol type classification in which mineral dust is part of the aerosol mixture."**

Page 7, lines 24-31: The authors need to add 12-month averages to Fig 6 so that the reader can follow along. None of the numbers in this paragraph correspond to the figure; presumably these numbers are averages over all 12 months, but the authors do not state this.

Thank you for this advice. These numbers do indeed refer to the annual average. We have added a new bar to the plots in Figure 6 that gives the annual average. We have also double-checked that the numbers in the text agree with Figure 6 and state that values refer to the annual average. As recommended by the other Referee, we have added information of the number of considered observations per month and year to provide further context for the fraction of aerosol types in %.

Page 8, line 33: This isn't really validation, as the authors do not have measurements to provide them with the "truth." Rather, this is basically a continuation of Section 3.2, but for different source regions.

**The Referee is correct, though the selected sites have been used in various studies to represent the respective aerosol types. We have replaced "validate" with "further test".**

I think the authors should include the seasonal variability at some of the sites in this section, similar to Fig 6. For instance, the West Africa has seasonal biomass burning as well as seasonal dust cycles. This should show up nicely at Banizoumbou and other African sites with a stacked chart like Fig 6.

Following the Referee's suggestion, we have investigated the potential of resolving seasonal variations in aerosol type using our typing scheme. We can indeed resolve changes in the occurrence of different aerosol types over West Africa based on the AERONET measurements at Dakar and Banizoumbou. We have added a respective paragraph in the discussion of the dust sites. However, we don't think that an additional figure is needed to transport this message. The new text is: "We have considered the AERONET sites Dakar and Banizoumbou to investigate if our methodology can be used to resolve the seasonal cycles of dust and biomass burning over West Africa (not shown). We find that the two sites are dominated by PD, DDM, and PDM but that the ratio of the three types varies with season. PD contributes strongest in spring (MAM, 71% at Dakar and 88% at Banizoumbou) and summer (JJA, 74% and 72%), goes down in autumn (SON, 55% and 63%), and has minimum contributions in winter (DJF, 37% and 49%). The decrease in PD comes with an increase in DDM from spring and summer to autumn and winter (Dakar: 25% in MAM, 21% in JJA, 40% in SON, and 45% in DJF; Banizoumbou: 11% in MAM, 23% in JJA, 36% in SON, and 40% in DJF). In addition, PDM has a maximum in winter (17% at Dakar and 10% at Banizoumbou) with values between 0% and 5% during the other seasons. Contributions of other aerosol types are negligible at the two sites throughout the year."

Figure 4: Why show PLDR instead of R\_p (from Eq 3)? If R\_p really does present the contribution of dust to the particle backscatter coefficient, it would be much more interesting to see R\_p than PLDR in this figure. Besides, PLDR only provides 4 broad categories of dust/pollution mixtures (dust, dust-dominated mixtures, pollution-dominated mixtures, and pollution), whereas the R\_p would provide much higher resolution.

Figure 4 and the respective discussion have been changed to focus on  $R_d$  (see reply to previous comment on use of PLDR vs  $R_d$ ).

Figure 5: I still don't like the multi-scale y-axis. Maybe try a single log scale for y-axis, if a single linear scale does not work.

Colors are difficult to discriminate, even for the non-color blind. Color-blind people have no chance of understanding this figure. I would add symbols to the lines for those folks.

And you don't really need all those error bars... May show one or more "representative" Error bars.... or just state some representative values in the caption. What do the error bars mean, anyways? 1-sigma? 2-sigma? 1 SDOM? 2 SDOM?

We are sorry for providing such a confusing figure. The figure has been revised for better visibility. Specifically, we now use a single linear y-axis, have removed all error bars (they were 1 SDOM), added symbols in the SSA plot and changed the lines to solid and dashed black and gray. We hope that these changes will make the figure more accessible also for colour blind readers.

Table 2: Something wrong here... FMF ranges from 0.31 to 0.90, but AE only ranges from 0.5 to 0.65. AE = 0.52 and 0.60 for PDM regions D&F, and AE=0.65 for polluted region G. All of these values are inconsistent with the literature, where AE < 1 is often used for "pure" dust. At a minimum, the authors should provide some discussion as to why their polluted sector has such a low AE.

We thank the Referee for spotting these inconsistencies. We did indeed use the wrong values for AE and SSA in Table 2. We have now included the proper number and double-checked that all parameters are consistent. We are sorry for this mistake.

Table 3: This is a little messy to look at. You don't really need the number of cases for each aerosol type, since that is covered by the first numerical column and the percentages. If you get rid of the number of cases, you'll have one value (the percentage) for each aerosol type, and this will be a lot easier for the reader to process.

Alternatively, this table could be converted into a stacked bar chart of percentages (like Fig 6). One stacked bar for each site.

Thank you for spotting this. The table has been revised to now only show the total number of cases and the percentages for each aerosol type.

**Minor issues:**

Page 2, line 28: replace "non-spherical" with "spherical."

We have corrected the typo.

[revised manuscript text omitted]

$$\delta_{\lambda}^{p} = \frac{1 - F_{22}(\lambda, 180^{\circ}) / F_{11}(\lambda, 180^{\circ})}{1 + F_{22}(\lambda, 180^{\circ}) / F_{11}(\lambda, 180^{\circ})}.$$
(2)

AERONET version 3 level 2.0 inversion products include PLDRs and lidar ratios at 440, 670, 870, and 1020 nm. The parameter sphericity, which was provided in the AERONET version 2 inversion product, has been discontinued in

25 version 3. We hence base this study on PLDR. This approach has the additional advantage that PLDR can also be obtained from independent measurements, e.g. with aerosol lidar. *Noh et al.* (2017) report that PLDR from the AERONET inversion products shows high correlation with lidar-derived values. PLDR can be used to calculate the contribution of dust to the particle backscatter coefficient for an external aerosol mixture. *Shimizu et al.* (2004) and *Tesche et al.* (2009) define the dust ratio ( $R_d$ ) as:

$$R_{\rm d} = \frac{(\delta^{\rm p} - \delta^{\rm p}_{\rm nd})(1 + \delta^{\rm p}_{\rm d})}{(\delta^{\rm p}_{\rm d} - \delta^{\rm p}_{\rm nd})(1 + \delta^{\rm p})}.$$
(3)

Here,  $\delta_{nd}^{p}$  and  $\delta_{d}^{p}$  indicate the PLDR of non-dust and pure dust particles, respectively. The corresponding values can be determined from lidar or AERONET observations (*Burton et al.*, 2014; *Burton et al.*, 2018). Shin et al. (2018) recently discussed AERONET-retrieved PLDR from different source regions. The authors find that PLDRs at 870 and 1020 nm are likely to be the two most reliable quantities. The authors based this finding on comparison to literature values obtained from lidar observa-

5 tions of pure dust particles which shows that AERONET-derived PLDRs at lower wavelengths are systematically lower than lidar observations at 532 nm. We accordingly apply the aerosol classification procedure to AERONET measurements at 1020 nm. We used values of  $\delta_{nd}^p = 0.02$  and  $\delta_d^p = 0.30$  for pure dust for Asian dust (*Shin et al.*, 2018). When PLDR was lower than  $\delta_{nd}^p$  or higher than  $\delta_d^p$ ,  $R_d$  was set to 0 or 1, respectively.

[revised manuscript text omitted]